# On the Identifiability of Poisson Branching Structural Causal Model Under Latent Confounding

Jie Qiao [* 1]  Zihuai Zeng [* 1]  Ruichu Cai [1 2]  Zhengming Chen [3]  Zhifeng Hao [3]

## Abstract

Causal discovery from observational count data poses unique challenges, particularly when the data exhibit inherent branching structures, such as an upstream ad impression event triggering a downstream purchase event with certain probability. Such branching dynamics are naturally modeled by thinning operators (for branching) and an independent Poisson distribution (for exogenous noise), constituting a Poisson Branching Structural Causal Model (PB-SCM). However, existing approaches based on PB-SCM rely on the restrictive assumption of causal sufficiency, failing to account for ubiquitous latent confounders. In this work, we propose a Latent Confounding Poisson Branching Structural Causal Model (LC-PB-SCM) to bridge this gap. We leverage Probability Generating Function (PGF) to characterize the complex dependencies introduced by latent confounding. Then, we establish a Trie representation theorem that maps the branching structure to algebraic properties of PGF monomials. Based on local PGF, we establish a complete identifiability condition for local 3-variables covering all causal patterns distinguishable up to monomial equivalence. Finally, we propose a practical algorithm to learn causal structures under latent confounding and demonstrate its effectiveness through experiments on both synthetic and real-world datasets.

## 1. Introduction

Causal discovery from observational count data is a fundamental challenge across various disciplines, ranging from

---

[*]Equal contribution [1]School of Computer Science, Guangdong University of Technology, Guangzhou, China [2]Peng Cheng Laboratory, Shenzhen, China [3]College of Mathematics and Computer, Shantou University, Shantou, China. Correspondence to: Ruichu Cai <cairuichu@gmail.com>.

*Proceedings of the 43rd International Conference on Machine Learning*, Seoul, South Korea. PMLR 306, 2026. Copyright 2026 by the author(s).

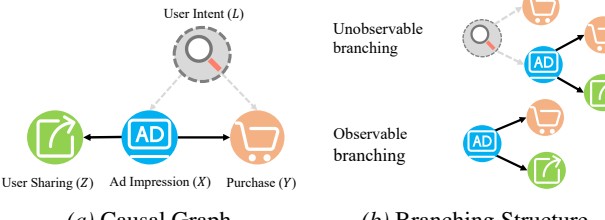

User Intent (*L*)

Unobservable branching

User Sharing (*Z*)   Ad Impression (*X*)   Purchase (*Y*)

Observable branching

*(a)* Causal Graph  *(b)* Branching Structure

*Figure 1.* (a) An example of the causal graph of count data in a digital advertising service. (b) An example showing the latent confounder $L$ affects both the ad impression $X$ and the purchase event $Y$ in a branching structure.

biology (Wiuf & Stumpf, 2006) to digital marketing (Weiß & Kim, 2014). Traditional methods for count data typically employ standard Bayesian networks (Park & Raskutti, 2015; 2017) or ordinal modeling (Ni & Mallick, 2022), which, however, treat counts merely as numerical values and overlook the inherent branching structure that is frequently encountered in many real-world applications (Weiß, 2018). Take Fig. 1 as an example, in the specific context of retargeting within digital advertising systems, a purchasing event is often directly inherited from an ad impression event. This relationship exhibits a branching structure modeled by the thinning operator '∘' (Steutel & van Harn, 1979) such that the purchase count $Y$ can be expressed as $Y = \alpha \circ X + \epsilon$, where $\alpha \circ X := \sum_{n=1}^{X} \xi_n^{(\alpha)}$ represents the successful conversions from $X$ impressions (with $\xi \sim \text{Bernoulli}(\alpha)$), and $\epsilon$ denotes exogenous noise. This formulation constitutes the core of the Poisson Branching Structural Causal Model (PB-SCM) (Qiao et al., 2024b; Xiang et al., 2024), offering a more physically meaningful representation that explicitly models these branching dynamics.

However, current approaches still heavily rely on the assumption of causal sufficiency, presuming that all common causes are observed. This assumption is rarely met in complex real-world systems where latent confounders are ubiquitous. For instance, as shown in Figure 1, unobserved factors that reflect user intention, such as their inaccessible search count on other platforms, often act as latent common causes, driving both the exposure to ads ($X$) and the final purchasing behavior ($Y$). In such cases, failure to account

for this latent confounding within the branching framework can lead to spurious causal conclusions and biased effect estimates.

Much effort has been made to address the challenges of latent confounders and branching structures modeling. In the presence of latent confounders, constraint-based methods such as FCI (Colombo et al., 2012; Spirtes et al., 2000) leverage conditional independence tests to output Partial Ancestral Graphs (PAGs). To go beyond Markov equivalence classes, a significant line of work introduces parametric assumptions. Early methods utilize rank conditions on covariance matrices to locate latent variables (Silva, 2006), while the Generalized Independent Noise (GIN) condition (Xie et al., 2020) extends this to linear non-Gaussian settings. This has inspired a series of subsequent works focusing on identifying linear (Xie et al., 2022; Huang et al., 2022; Chen et al., 2023) and nonlinear (Kong et al., 2023) latent hierarchical structures. Recently, these rank-based ideas have also been extended to discrete data via tensor rank decompositions (Chen et al., 2024; 2025), alongside the emergence of scalable differentiable methods for both ADMG learning (Ma et al., 2024) and latent hierarchical discovery (Prashant et al., 2025). However, these approaches remain inadequate for causal discovery in count data due to the overlook of the inherent branching structure. For branching structure modeling, the core mechanism relies on the thinning operator (Steutel & van Harn, 1979), classically used in integer-valued autoregressive (INAR) models (Weiß, 2018; Al-Osh & Alzaid, 1987; McKenzie, 1985). By incorporating this operator to capture causal dynamics, the PB-SCM framework (Qiao et al., 2024b; Xiang et al., 2024) offers an explicit representation of branching processes. However, current approaches remain limited as they rely on the strict assumption of causal sufficiency. Consequently, it remains unclear how to learn causal structures in the presence of latent confounding.

In this work, we propose the **L**atent **C**onfounding **P**oisson **B**ranching **S**tructural **C**ausal **M**odel (LC-PB-SCM) to bridge this gap. We leverage the Probability Generating Function (PGF) to characterize the complex dependencies in the data. We focus on scenarios where latent confounders are exogenous, which simplifies the topological complexity while preserving the capability to model the most common confounding effects in practical applications. To tackle the identification challenge when latent confounders exist, we introduce a method based on local PGFs, which allows us to inspect specific substructures of the graph. We observe that the branching structure induces a unique algebraic form in the PGF, which can be captured by a novel graphical representation *Trie*. This tree-like structure elucidates the intrinsic connection between local causal mechanisms and the monomials in the PGF expansion.

By mapping every branching structure to a Trie, we establish a rigorous link between causal edges and the corresponding monomials in the logarithm of PGF, which can determine the presence of causal edges and latent confounders by examining these monomials. Based on this theoretical insight, we derive various identifiability conditions and demonstrate that the proposed identifiability condition within local 3-variable substructures is complete. Finally, we introduce a practical algorithm for recovering causal structures from observational count data. Our main contributions are summarized as follows:

- We introduce Trie, a novel graphical representation that captures the branching relationships and maps them to the algebraic properties of the PGF.

- We investigate the identifiability of LC-PB-SCM up to monomial equivalence and demonstrate that the proposed identification condition based on local PGFs is complete for local 3-variable scenarios.

- We propose a practical algorithm to learn the causal structure in the presence of latent confounders and demonstrate the effectiveness through experiments on both synthetic and real-world datasets.

## 2. Latent Confounding Poisson Branching Structural Causal Model

In this section, we first introduce the Latent Confounding Poisson Branching Structural Causal Model (LC-PB-SCM) and the graphical representation used to identify the causal structure. Then, we present the necessary preliminaries regarding Probability Generating Functions (PGF).

### 2.1. Problem Formulation

We now formally define our model and our goal. Let $\mathbf{X} = \{X_1, \ldots, X_{d+r}\}$ represent a multi-dimensional random Poisson count, whose causal structure is given by a Directed Acyclic Graph (DAG) $G(\mathbf{V}, \mathbf{E})$. Here, the vertex set $\mathbf{V} = \mathbf{O} \cup \mathbf{L}$ is partitioned into observed variables $\mathbf{O} = \{1, \ldots, d\}$ and latent confounders $\mathbf{L} = \{d+1, \ldots, d+r\}$, and $\mathbf{E}$ denotes the edge set. For symbol simplicity, we use $L_i$ to denote the latent confounder $X_{d+i}$ where $(d+i) \in \mathbf{L}$. We use $Pa(i) = \{j \mid j \to i \in \mathbf{E}\}$, $Ch(i) = \{j \mid i \to j \in \mathbf{E}\}$, and $Des(i) = \{j \mid i \rightsquigarrow j \in \mathbf{E}\}$ denote the sets of parents, children and descendants of vertex $i$ in $\mathcal{G}(\mathbf{V}, \mathbf{E})$. Moreover, we define a *directed path* $p = (i_1, \ldots, i_n)$ in $\mathcal{G}$ as a sequence of vertices of $\mathcal{G}$ where there is a directed edge from $i_j$ to $i_{j+1}$ for $1 \le j \le n-1$. We assume that any variable in $\mathbf{X}$ satisfies the following Latent Confounding Poisson Branching Structural Causal Model:

**Definition 2.1** (Latent Confounding Poisson Branching Structural Causal Model)**.** For the set of observed variables $\mathbf{X_O}$ and latent confounders $\mathbf{X_L}$, let $\epsilon_i \sim \text{Poisson}(\mu_i)$ be

the independent exogenous noise. Each observed variable $X_i \in \mathbf{X_O}$ is generated by:

$$X_i = \sum_{j \in Pa_{\mathbf{O}}(i)} \alpha_{j,i} \circ X_j + \sum_{k \in Pa_{\mathbf{L}}(i)} \alpha_{k,i} \circ X_k + \epsilon_i, \quad (1)$$

where $\alpha_{j,i}, \alpha_{k,i} \in (0,1)$ are the causal strengths from observed parent $j$ and latent parent $k$ to $i$, respectively. The latent confounders are mutually independent roots with $X_k \sim \text{Poisson}(\mu_k)$. The operator '$\circ$' denotes Binomial thinning such that $\alpha \circ Y = \sum_{n=1}^{Y} \xi_n^{(\alpha)}$, where $\xi_n^{(\alpha)} \overset{\text{i.i.d.}}{\sim} \text{Bernoulli}(\alpha)$, independent of $Y$.

This formulation extends standard Poisson branching models (Qiao et al., 2024a; Al-Osh & Alzaid, 1987) by incorporating latent confounders. We further assume the standard *faithfulness* and *causal Markov* assumptions hold for the underlying DAG $\mathcal{G}$.

In this work, we focus explicitly on *exogenous latent confounding* and exclude intermediate latent mediators (e.g., $X_i \to L \to X_j$) from our discussion. Due to the compositional property of the binomial thinning operator, such structures are distributionally equivalent to a direct edge $X_i \to X_j$ with a scaled coefficient (see Appendix F). While this equivalence may not generally hold in more complex graph topologies with interactive effects, we restrict our focus here to the primary confounding paths and assume that all latent variables are root nodes.

Furthermore, since the latent variables $\mathbf{L}$ are unobserved, the full DAG $\mathcal{G}$ is generally not identifiable. Instead, our goal is to recover the causal relations among the observed variables $\mathbf{O}$, while accounting for the correlations induced by $\mathbf{L}$. To represent this *marginalized* causal structure in LC-PB-SCM, we employ a special type of Acyclic Directed Mixed Graph (ADMG) (Ashman et al., 2023; Richardson & Spirtes, 2002), called Parental Acyclic Directed Mixed Graph (PADMG).

**Definition 2.2** (Parental Acyclic Directed Mixed Graph). A Parental Acyclic Directed Mixed Graph is an acyclic directed mixed graph $\mathcal{G}_{\mathbf{O}} = (\mathbf{O}, \boldsymbol{E}_O)$ defined over the observed variables, where the edge set $\boldsymbol{E}_O$ contains directed edges ($\to$) and bi-directed edges ($\leftrightarrow$). The graph satisfies the following properties:

- **Parental Relations:** A directed edge $i \to j$ exists if and only if $i$ is a direct parent of $j$ in the underlying DAG (i.e., $i \in Pa_{\mathbf{O}}(j)$).

- **Latent Confounding:** A bi-directed edge $i \leftrightarrow j$ exists if and only if $i$ and $j$ share a latent parent in $\mathbf{L}$ (i.e., $Pa_{\mathbf{L}}(i) \cap Pa_{\mathbf{L}}(j) \neq \varnothing$).

- **Acyclicity:** The graph contains no directed cycles.

Note that the main difference between PADMG and ADMG

is that ADMG is designed to encode the conditional independencies and the ancestral relation; PADMG strictly encodes direct parent-child relationships. This formulation is essential for LC-PB-SCM. Unlike the linear functional relation, marginalization of the discrete and non-linear nature of the binomial thinning operator would obscure the specific functional form required for identifiability (see Appendix F for details).

Based on the PADMG, we can now define the graphical representation of the distributional equivalence of LC-PB-SCM, as the true PADMG may not be uniquely identifiable from the observational distribution $P(\mathbf{X})$. Instead, we can only identify an equivalence class of graphs that are statistically indistinguishable. To represent this equivalence class, we introduce the Partial Parental Acyclic Directed Mixed Graph (PPADMG).

**Definition 2.3** (Partial Parental Acyclic Directed Mixed Graph). Let $[\mathcal{G}_{\mathbf{O}}]$ denote the equivalence class of all Parental Acyclic Directed Mixed Graphs that are distributionally equivalent to the true structure. A Partial Parental Acyclic Directed Mixed Graph (PPADMG), denoted as $\mathcal{P}$, is a graphical representation of $[\mathcal{G}_{\mathbf{O}}]$ that has the same skeleton as any graph in $[\mathcal{G}_{\mathbf{O}}]$. It consists of the same vertex set $\mathbf{O}$ and edges with three types of endpoints: tail ($-$), arrowhead ($>$), and circle ($\circ$) such that

- A *tail* at $i$ (e.g., $i \to j$) appears if $i$ is a parent of $j$ in every graph in $[\mathcal{G}_{\mathbf{O}}]$.

- An *arrowhead* at $i$ (e.g., $i \leftarrow j$ or $i \leftrightarrow j$) appears if $i$ is not a parent of $j$ in every graph in $[\mathcal{G}_{\mathbf{O}}]$.

- A *circle* at $i$ (e.g., $i \circ\!\!\to j$) appears if there exists at least one graph in $[\mathcal{G}_{\mathbf{O}}]$ where $i$ is a parent of $j$, and at least one graph where $i$ is not a parent of $j$ (indicating uncertainty about the direct causal direction).

- A *bow structure* appears at $i$ and $j$ (e.g., $i \leftarrow\!\!- k -\!\!\to j$ together with $i \to j$) if there is latent confounder $k$ affecting both $i, j$ and $i$ is a directed parent of $j$.

Note that, in PPADMG, we include a type of bow structure to represent a confounded edge that is often considered infeasible. In LC-PB-SCM (will be discussed in the following section), this type of structure is feasible, and thus, we include it explicitly in our graphical representation.

**Goal.** Given $m$ i.i.d. samples $\mathcal{D} = \{\mathbf{x}^{(l)}\}_{l=1}^{m}$ generated by an unknown LC-PB-SCM, our objective is to learn the PPADMG $\mathcal{P}$ that represents the equivalence class of the underlying causal structure among the observed variables.

### 2.2. Preliminary

The Probability Generating Function (PGF) is a powerful tool for analyzing discrete random variables, uniquely en-

coding their distributions (Winner & Sheldon, 2016).

**Definition 2.4** (Probability Generating Function). Let $\mathbf{X} = (X_1, \ldots, X_d)$ be a vector of non-negative integer-valued random variables. Let $\mathbf{s} = (s_1, \ldots, s_d)$ be a vector of auxiliary indeterminates. The joint PGF of $\mathbf{X}$, denoted as $G_{\mathbf{X}}(\mathbf{s})$, is defined as:

$$G_{\mathbf{X}}(\mathbf{s}) = \mathbb{E}\left[\prod_{i=1}^d s_i^{x_i}\right] = \sum_{x_1, \ldots, x_d} P(x_1, \ldots, x_d) \prod_{i=1}^d s_i^{x_i}. \tag{2}$$

Since the full joint PGF is a complex composite function (see Theorem A.2, Eq. A.3–A.4), fully expanding it is intractable for estimation. We further introduce the Local Probability Generating Function, which allows us to focus on the local structure.

**Definition 2.5** (Local Probability Generating Function). Given the joint PGF $G_{\mathbf{X}}(\mathbf{s})$ of $\mathbf{X} = (X_1, \ldots, X_d)$, the local PGF of vertices $\mathbf{L} \subset [d]$ is obtained by setting $\mathbf{s}_{[d] \setminus \mathbf{L}}$ approach zero such that $G_{\mathbf{X}}^{\mathbf{L}}(\mathbf{s}) = \lim_{\mathbf{s}_{[d] \setminus \mathbf{L}} \to \mathbf{0}} G_{\mathbf{X}}(\mathbf{s})$.

Intuitively, the local PGF is designed to simplify the analysis by focusing on local structures. Thus, by setting the non-target variables to zero, we retain only the monomials involving the target variables to isolate the local causal structure of interest, where a limiting process is used to ensure mathematical rigor.

Moreover, one key property of PGFs in deriving our model's identifiability conditions is the marginalization, which allows us to easily analyze the PGF of observed variables by marginalizing out latent confounders.

**Property 2.6** (Marginalization). Given the joint PGF $G_{\mathbf{X}}(\mathbf{s})$ of $\mathbf{X}$, the PGF of a marginal subset of variables $\mathbf{X}_{\mathcal{A}}$ (where $\mathcal{A} \subset \{1, \ldots, d\}$) is obtained by setting $s_j = 1$ for all $j \notin \mathcal{A}$:

$$G_{\mathbf{X}_{\mathcal{A}}}(\mathbf{s}_{\mathcal{A}}) = G_{\mathbf{X}}(\mathbf{s}_{\mathcal{A}}, \mathbf{s}_{\mathcal{A}^c} = \mathbf{1}). \tag{3}$$

## 3. Identifiability

In this section, we systematically investigate the identifiability of the Latent Confounding Poisson Branching Structural Causal Model. We begin with a motivating example to demonstrate the use of the Trie, a novel graphical representation that links causal structure to the algebraic form of the Log-Probability Generating Function (log-PGF). This connection enables us to derive complete identifiability rules for local structures, even in the presence of latent confounders.

### 3.1. Motivating Example

Before detailing our theoretical results, we present a motivating example to illustrate the core intuition behind our approach. Consider the Instrumental Variable (IV) scenarios shown in Figure 2, where we aim to identify the causal

direction (highlighted in red) between $X_3$ and $X_1$ in the presence of a latent confounder $L$ affecting both $X_1$ and $X_2$. Note that we use $X_3 \circ\!\!\rightarrow X_1$ to represent the structural uncertainty within the equivalence class regarding the instrument $X_3$.

Identifying such an instrument variable can be challenging due to the unobserved confounding induced by $L$. However, we demonstrate that by analyzing the logarithmic Probability Generating Function (log-PGF) of the observed variables, asymmetry emerges that renders these graphs identifiable.

Our key insight links the expansion of the marginal log-PGF directly to the graph topology. As illustrated in Fig. 2(a) and (b), the direct paths originating from any root node can be merged based on shared prefixes to form a prefix tree, which we define as a Trie. Each Trie or Pruned Trie corresponds to a specific term in the log-PGF expansion. For instance, $X_1 \to X_2$ generates the term $\alpha_{1,2} s_1 s_2$. We term the variable component (e.g., $s_1 s_2$) a *monomial*, which serves as a structural fingerprint. Consequently, the total log-PGF is essentially the aggregation of these monomials across all root nodes.

By enumerating the monomial set $\mathcal{M}(\mathcal{G})$ for both structures, we observe that even though the latent confounder $L$ is unobserved (implying $s_L = 1$ after marginalization), the monomial $s_1 s_2^2 s_3$ appears exclusively in $\mathcal{G}_2$. As illustrated in Fig. 2(b), this monomial corresponds to a Trie generated by merging shared prefixes from the paths originating at $L$, which is impossible to construct in $\mathcal{G}_1$. This discrepancy implies that $\mathcal{G}_1$ and $\mathcal{G}_2$ are distributionally distinguishable, as no non-degenerate coefficient exists that can render their PGFs identical. This motivates our use of monomial analysis to identify causal structures in the presence of latent confounding. The subsequent sections will generalize this observation, based on the property of local-PGF, proving rule completeness for local 3-node structures and establishing a global identification algorithm.

### 3.2. Trie-based Graphical Representation

To systematically investigate the identifiability problem in the presence of latent confounding, this section formally introduces the Trie-based graphical representation, establishing a connection between the Trie and the monomial in the log-PGF. We begin with the definition of Trie.

**Definition 3.1** (Trie). A Trie rooted at $v_0 \in \mathbf{V}$ over $\mathcal{G}(\mathbf{V}, \mathbf{E})$ denoted as $\mathcal{T}_G(v_0) = (\mathbf{N}, \mathbf{E}_{\mathcal{T}})$, is a tree defined as follows:

- The set of nodes $\mathbf{N}$ is *induced by* the set of all valid directed paths originating from $v_0$ in $\mathcal{G}$. That is, for every valid directed path $\pi$ starting at $v_0$, there exists a unique corresponding node $n_\pi \in \mathbf{N}$.

- A directed edge $(n_\pi, n_{\pi'})$ exists in $\mathbf{E}_{\mathcal{T}}$ if and only if $\pi'$ is obtained by appending a single vertex $v$ to $\pi$ such

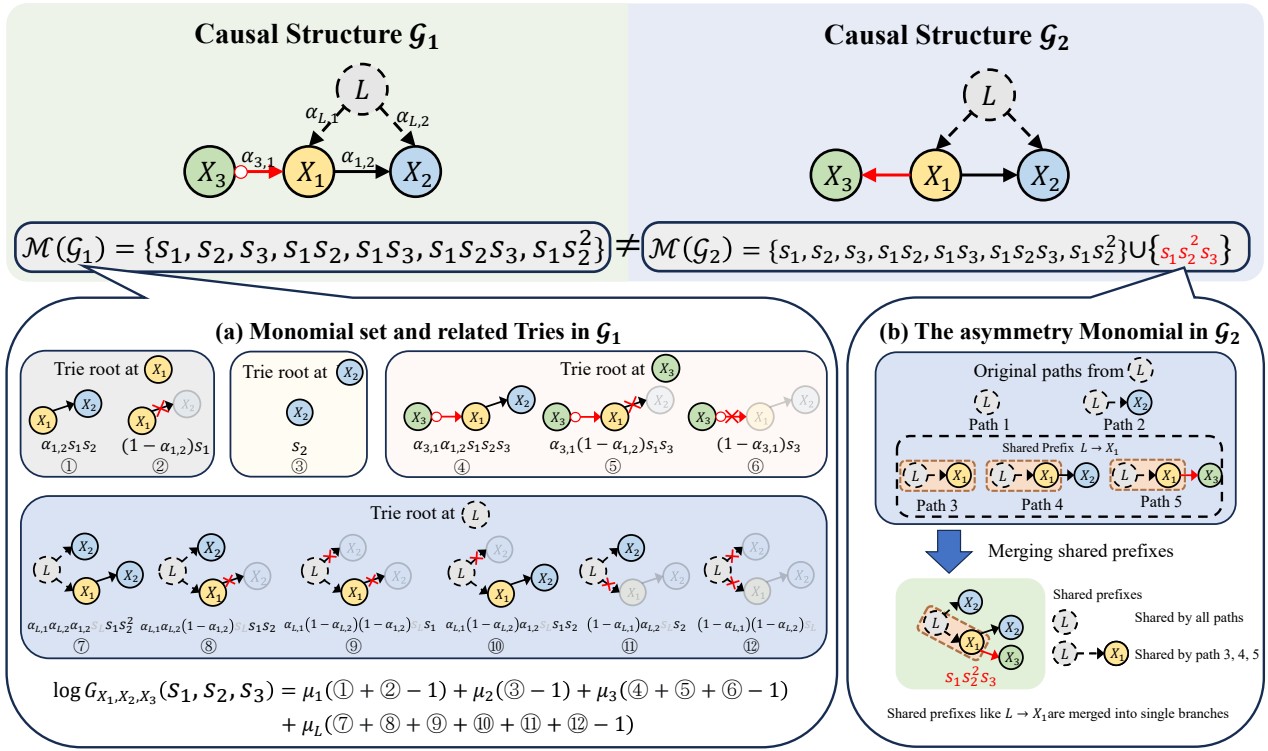

*Figure 2.* An example of constructing a Trie structure and the illustration of the asymmetry of two causal graphs using the monomial set.

that $(last(\pi), v) \in \mathbf{E}$. We denote this as $\pi' = \pi \oplus v$, where $\oplus$ represents the path concatenation operator.

Essentially, a Trie is obtained by merging a prefix of each direct path from some root. Take Fig. 2(b) as an example, a Trie root at $L$ over $\mathcal{G}_2$ consists of five valid directed paths. Since all nodes in this tree are induced by directed paths, we have five nodes in this tree. Moreover, the directed path with shared prefixes, e.g., $L \to X_1$, is shared by path 3, path 4, and path 5, and will be merged. Intuitively, this structure captures how Poisson counts propagate through the network. In addition, the core intuition behind defining a Trie node by its entire directed path $\pi$ (denoted $n_\pi$) instead of just the graph vertex is to ensure uniqueness, so that the same vertex may appear multiple times in a Trie when it is reachable via different directed paths from the root. For example, considering Trie $\mathcal{T}_{\mathcal{G}_2}(L)$ shown in Fig. 2(b), vertex $X_2$ induces two distinct Trie nodes including $n_{(L,X_2)}$ (induced by path $L \to X_2$) and $n_{(L,X_1,X_2)}$ (induced by path $L \to X_1 \to X_2$). This property is a direct consequence of the PGF of PB-SCM, as it is a composition of multiple directed paths. However, in the physical process of Binomial thinning, a connection may either transmit a count (active) or fail to transmit (inactive), which is naturally represented by a *Pruned Trie*.

**Definition 3.2** (Pruned Trie). Let $\mathcal{T}_{\mathcal{G}}(v_0) = (\mathbf{N}, \mathbf{E}_{\mathcal{T}})$ be the original Trie and $\mathcal{E}_{cut} \subset \mathbf{E}_{\mathcal{T}}$ be the subset of edges to

be pruned. The Pruned Trie $\mathcal{T}' = (\mathbf{N}', \mathbf{E}')$ of Trie $\mathcal{T}_{\mathcal{G}}(v_0)$ is defined as the connected subtree containing the root $n_{v_0}$ within the subgraph $(\mathbf{N}, \mathbf{E}_{\mathcal{T}} \setminus \mathcal{E}_{cut})$, where the node set $\mathbf{N}' \subseteq \mathbf{N}$ consists only those nodes $n_\pi$ that remain reachable from the root through the remaining edges, while $\mathbf{E}' = \{(u, v) \in \mathbf{E}_{\mathcal{T}} \setminus \mathcal{E}_{cut} \mid u \in \mathbf{N}'\}$ denotes the set of edges connecting these remaining nodes.

A Pruned Trie is essentially a tree pruned by a pruning set such that cutting an edge implies cutting all its descendants. For example, in Fig. 2(a), if edge $L \to X_1$ is cut in ⑪, its descendant is also cut. However, a pruning set might be redundant, and thus, to ensure compactness, we define the Minimal Pruning Set:

**Definition 3.3** (Minimal Pruning Set). A pruning set $\mathcal{E}_{cut}^* \subset \mathbf{E}_{\mathcal{T}}$ is minimal if it contains no redundant edges such that for every edge $(u, v) \in \mathcal{E}_{cut}^*$, the unique path from the root $n_{v_0}$ to $u$ in $\mathcal{T}_{\mathcal{G}}(v_0)$ contains no edges belonging to $\mathcal{E}_{cut}^*$.

This leads to the definition of the complete search space for our structural identification through edge removal.

**Definition 3.4** (Space of Pruned Tries). Let $\mathcal{T}_{\mathcal{G}}(v_0)$ be the original Trie. The space of all possible Pruned Tries, denoted as $\mathbb{S}(\mathcal{T}_{\mathcal{G}}(v_0))$, is defined as the set of all unique subtrees $\mathcal{T}'$ that can be obtained by applying any pruning set $\mathcal{E}_{cut} \subseteq \mathbf{E}_{\mathcal{T}}$ to $\mathcal{T}_{\mathcal{G}}(v_0)$ with empty set in $\varnothing \in \mathcal{E}_{cut}$ such that $\mathbb{S}(\mathcal{T}_{\mathcal{G}}(v_0)) = \{\mathcal{T}' \mid \text{for any } \mathcal{E}_{cut} \subseteq \mathbf{E}_{\mathcal{T}}\}$.

Essentially, $\mathbb{S}(\mathcal{T}_{\mathcal{G}}(v_0))$ enumerates every possible sub-tree that can be obtained by selecting any subset of edges $\mathcal{E}_{cut} \subseteq \mathbf{E}_{\mathcal{T}}$ to prune such that $\mathbb{S}(\mathcal{T}_{\mathcal{G}}(v_0)) = \{\mathcal{T}' | \text{for any } \mathcal{E}_{cut} \subseteq \mathbf{E}_{\mathcal{T}}\}$ where $\mathcal{T}'$ is the corresponding Pruned Trie by each $\mathcal{E}_{cut}$. The intent of Definition 3.4 is to define the complete search space over all Pruned Tries derivable from the original Trie $\mathcal{T}_G(v_0)$.

We now present our main theorem, which establishes the closed-form relationship between Trie and log-PGF. This theorem validates the intuition discussed in the motivating example that every term in the log-PGF corresponds exactly to a specific Pruned Trie.

**Theorem 3.5** (Monomial Representation Theorem). *Let* $\mathbf{X}$ *be a random vector generated by an LC-PB-SCM with causal graph* $\mathcal{G}$. *Then the log probability generating function (log-PGF) of* $\mathbf{X}$ *can be expressed as:*

$$\log G_{\mathbf{X}}(\mathbf{s}) = \sum_{k \in \mathbf{V}} \mu_k \left[ \sum_{\mathcal{T}^* \in \mathbb{S}(\mathcal{T}_{\mathcal{G}}(v_k))} c(\mathcal{T}^*) m(\mathcal{T}^*) - 1 \right] \quad (4)$$

*where* $\mathcal{T}^*$ *is the Pruned Trie in the space with node set* $\mathbf{N}_{\mathcal{T}}^*$ *and the minimal pruning set* $\mathcal{E}_{cut}^*$, $m(\mathcal{T}^*) := \prod_{j \in \mathbf{N}_{\mathcal{T}}^*} s_j$ *is corresponding monomial represented by the Pruned Trie and* $c(\mathcal{T}^*) := \prod_{(u,v) \in \mathbf{E}_{\mathcal{T}}^*} \alpha_{u,v} \cdot \prod_{(u,v) \in \mathcal{E}_{cut}^*} (1 - \alpha_{u,v})$ *with* $c(\mathcal{T}^*) = 1$ *when* $|\mathbf{N}_{\mathcal{T}}^*| = 1$, *is the coefficient of the monomial.*

Theorem 3.5 shows that a log-PGF can be expressed by enumerating the Pruned Trie and the monomial, and its coefficient can be deduced by Trie using $m(\mathcal{T}^*)$ and $c(\mathcal{T}^*)$, respectively. Based on this theorem, we can formally define the condition under which two causal graphs are distinguishable via their PGFs.

**Definition 3.6** (Monomial Equivalence). *Let* $\mathcal{M}(\mathcal{G})$ *be the set of monomials appearing in the log-probability generating function (log-PGF) associated with an LC-PB-SCM over a DAG* $\mathcal{G}$ *such that* $\mathcal{M}(\mathcal{G}) = \{m(\mathcal{T}') \mid \mathcal{T}' \in \mathbb{S}(\mathcal{T}_{\mathcal{G}}(v_k)), k \in \mathbf{V}\}$ *where* $m(\mathcal{T}')$ *denotes the corresponding monomial as define at theorem 3.5. Two causal graphs* $\mathcal{G}_1$ *and* $\mathcal{G}_2$ *are said to be monomial equivalent if and only if they have the identical set of monomials, i.e.,* $\mathcal{M}(\mathcal{G}_1) = \mathcal{M}(\mathcal{G}_2)$.

This definition implies that if two graphs are not monomial Equivalent, they are distributionally distinguishable as there do not exist any coefficients to make the PGF identical. Therefore, the identifiability can be investigated through the analysis of the Trie structure, as it has a one-to-one correspondence to the monomial.

In the following section, we will utilize this Trie-based analysis to prove the identifiability rules for 3-node structures using local PGF properties.

### 3.3. Identifiability of Local Structure

This subsection establishes the monomial conditions for identifying the monomial equivalence class represented by PPADMG. To show the identifiability of LC-PB-SCM, we utilize the local PGF given in Definition 2.5. This property is attractive as it allows us to isolate and examine some specific components by strategically letting the other variables $\mathbf{s}$ approach zero within the generating function. This allows us to investigate the local structure and devise a practical structure learning algorithm. Therefore, we focus on local structures involving up to three observed variables. A comprehensive summary of these rules is given in Table 3. Our identification results begin with the most fundamental building blocks: determining the existence of edges (skeleton) and detecting latent confounding on adjacency edges.

**Theorem 3.7** (Identifiability of Skeleton). *Let* $X_1, X_2 \in \mathbf{X_O}$ *be two arbitrary observed variables in LC-PB-SCM under causal graph* $\mathcal{G}$. $X_1$ *and* $X_2$ *are adjacent in PPADMG* $\mathcal{P}$ *if and only if the monomial* $s_1 s_2 \in \mathcal{M}(\mathcal{G})$.

Theorem 3.7 establishes the identifiability of the causal skeleton. Note that without additional information, the adjacent variables will fall into an equivalence class, and thus the two variable cases are unidentifiable (see Appendix B.2 for the proof). To break this symmetry, we first leverage the higher-order monomials such as $s_1 s_2^2$, which leads to the following theorem:

**Theorem 3.8** (Identifiability of Confounded Edge). *Let* $X_1, X_2 \in \mathbf{X_O}$ *be two arbitrary observed variables in LC-PB-SCM under causal graph* $\mathcal{G}$. *Given the adjacent causal pair* $X_1 \circ\!\!-\!\!\circ X_2$ *in PPADMG* $\mathcal{P}$, *the causal structure of* $X_1 \to X_2$ *with latent confounder* $L$ *affecting both* $X_1$ *and* $X_2$ *holds if and only if the monomial* $s_1 s_2^2 \in \mathcal{M}(\mathcal{G})$.

Once the skeleton and direct confounding are established, the next challenge is orienting the edges. One critical subclass of local structures involves colliders. In our framework, colliders generate unique interaction terms that enable us to distinguish them from chains or forks, even under latent confounding.

**Theorem 3.9** (Identifiability of Collider Structure). *Let* $X_1, X_2, X_3 \in \mathbf{X_O}$ *be three arbitrary observed variables in LC-PB-SCM under causal graph* $\mathcal{G}$. *For the structures in Figures 3(c1) and 3(c2), let the non-target edges constitute the structural context. Given the structural context, the target collider edges are identifiable under the following monomial condition:*

- ***Type (c1):*** *if* $s_1 s_2 s_3^2 \in \mathcal{M}(\mathcal{G})$, *then the target edges* $X_1 \circ\!\!\rightarrow X_3$ *and* $X_2 \circ\!\!\rightarrow X_3$ *are identifiable;*

- ***Type (c2):*** *if* $s_1 s_2 s_3 \notin \mathcal{M}(\mathcal{G})$, *then the target edges* $X_1 \circ\!\!\rightarrow X_3$ *and* $X_2 \circ\!\!\rightarrow X_3$ *are identifiable.*

Moreover, our approach can further distinguish causal direc-

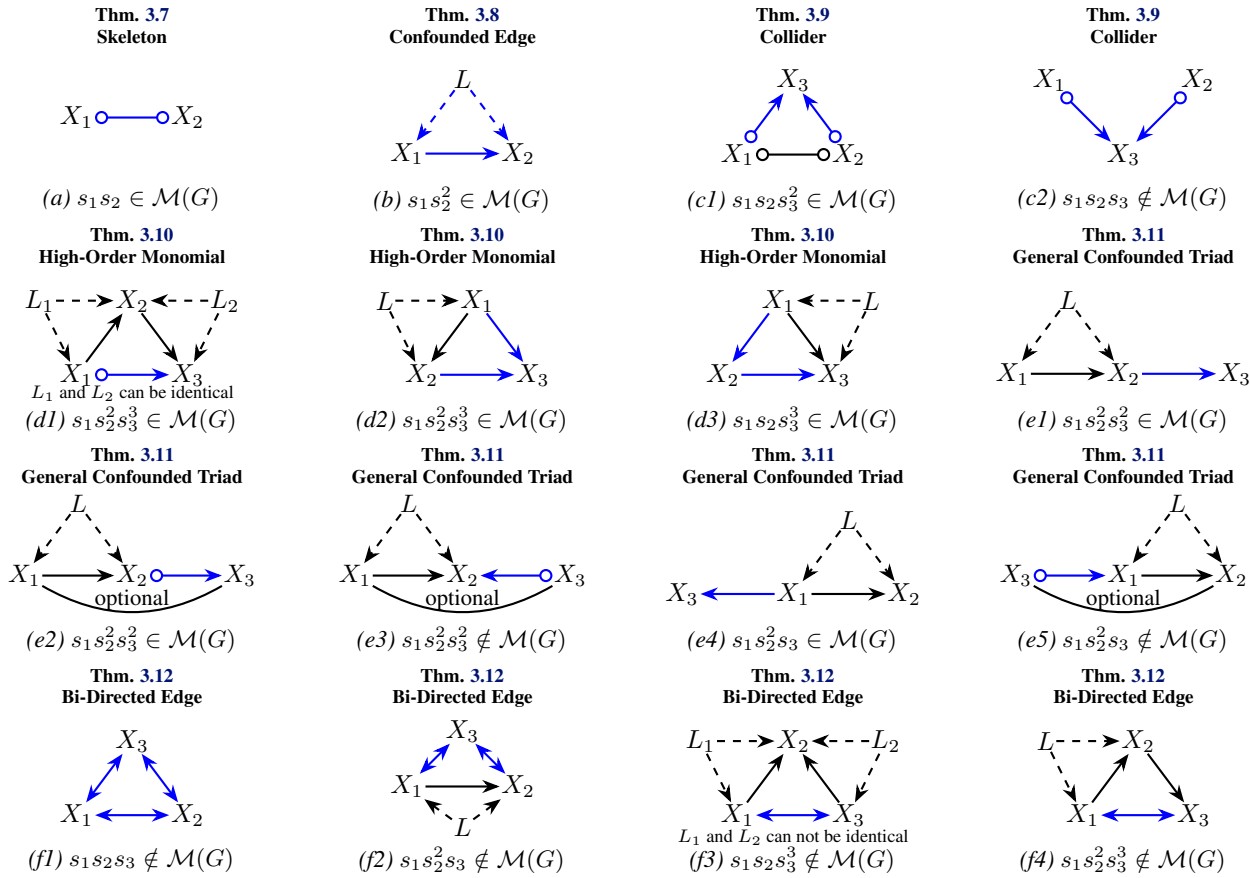

*Figure 3.* Identifiability of target causal relations (blue edges) in LC-PB-SCM. Monomial conditions are specified in the subfigure titles.

tions by analyzing higher-order monomials within 3-node substructures.

**Theorem 3.10** (Identifiability of High-Order Monomial). *Let $X_1, X_2, X_3 \in \mathbf{X_O}$ be three arbitrary observed variables in LC-PB-SCM under causal graph $\mathcal{G}$. For the structures in Figures 3(d1) to 3(d3), let the non-target edges constitute the structural context. Given the structural context, the target edges are identifiable under the following monomial condition:*

- ***Type (d1):*** *if $s_1 s_2^2 s_3^3 \in \mathcal{M}(\mathcal{G})$ holds, then the target edge $X_1 \circ\!\!\rightarrow X_3$ is identifiable, where the two latent variables are not necessarily distinct.*

- ***Type (d2):*** *if $s_1 s_2^2 s_3^3 \in \mathcal{M}(\mathcal{G})$ holds, then the target edges $X_1 \rightarrow X_3$ and $X_2 \rightarrow X_3$ are identifiable.*

- ***Type (d3):*** *if $s_1 s_2 s_3^3 \in \mathcal{M}(\mathcal{G})$ holds, then the target edges $X_1 \circ\!\!\rightarrow X_2$ and $X_2 \circ\!\!\rightarrow X_3$ are identifiable.*

Moreover, we address the general variations in complex triads, which require specific monomial checks to resolve the edge orientations fully.

**Theorem 3.11** (General Confounded Triad). *Let $X_1, X_2, X_3 \in \mathbf{X_O}$ be three arbitrary observed variables in*

*LC-PB-SCM under causal graph $\mathcal{G}$. For the structures in Figures 3(e1) to 3(e5), let the non-target edges constitute the structural context. Given the structural context, the target edges are identifiable under the following monomial condition:*

- ***Type (e1):*** *if $s_1 s_2^2 s_3^2 \in \mathcal{M}(\mathcal{G})$ holds, then the target edges $X_2 \rightarrow X_3$ is identifiable;*

- ***Type (e2):*** *if $s_1 s_2^2 s_3^2 \in \mathcal{M}(\mathcal{G})$ holds, then the target edges $X_2 \circ\!\!\rightarrow X_3$ is identifiable;*

- ***Type (e3):*** *if $s_1 s_2^2 s_3^2 \notin \mathcal{M}(\mathcal{G})$ holds, then the target edges $X_3 \circ\!\!\rightarrow X_2$ is identifiable;*

- ***Type (e4):*** *if $s_1 s_2^2 s_3 \in \mathcal{M}(\mathcal{G})$ holds, then the target edges $X_1 \rightarrow X_3$ is identifiable;*

- ***Type (e5):*** *if $s_1 s_2^2 s_3 \notin \mathcal{M}(\mathcal{G})$ holds, then the target edges $X_3 \circ\!\!\rightarrow X_1$ is identifiable.*

Finally, since the presence of latent confounders introduces bi-directed edges indicating the existence of a latent confounder, in the following theorem, we examine monomials that capture the specific signature of such a shared latent parent.

**Algorithm 1:** Causal Discovery for LC-PB-SCM

---

**Input:** Count Data $\mathcal{D}$
**Output:** PPADMG $\mathcal{P}$
Initialize $\mathcal{P}$ with vertices **V**.
// Phase 1:  Skeleton Identification
**foreach** *pair* $(X_i, X_j)$ **do**
    **if** $Test(\frac{\partial^2 \log \hat{G}_{\mathbf{X}}^{\{i,j\}}(\mathbf{s})}{\partial s_i \partial s_j} \neq 0))$ **then**
        Add edge $X_i - X_j$

// Phase 2:  Latent Confounded Edge
**foreach** *adjacent pairs in* $\mathcal{P}$ **do**
    **if** $Test(\frac{\partial^3 \log \hat{G}_{\mathbf{X}}^{\{i,j\}}(\mathbf{s})}{\partial s_i \partial s_j^2} \neq 0))$ **then**
        Orient $X_i \rightarrow X_j$ together with
          $X_i \leftarrow\text{-- } L \text{ --}\rightarrow X_j$

// Phase 3:  Triples Orientation
**foreach** *connected triples in* $\mathcal{P}$ **do**
    Orient edges based on Theorems 3.9 to 3.12;
**return** $\mathcal{P}$

---

**Theorem 3.12** (Identifiability of Bi-directed Edge Structure). *Let $X_1, X_2, X_3 \in \mathbf{X_O}$ be three arbitrary observed variables in LC-PB-SCM under causal graph $\mathcal{G}$. For the structures in Figures 3(f1) to 3(f4), let the non-target edges constitute the structural context. Given the structural context, the target bi-directed edges are identifiable under the following monomial condition:*

- ***Type (f1):*** *if $s_1 s_2 s_3 \notin \mathcal{M}(\mathcal{G})$ holds, then the target edges $X_1 \leftrightarrow X_2$, $X_2 \leftrightarrow X_3$, and $X_1 \leftrightarrow X_3$ are identifiable;*

- ***Type (f2):*** *if $s_1 s_2^2 s_3^3 \notin \mathcal{M}(\mathcal{G})$ holds, then the target edges $X_1 \rightarrow X_3$ and $X_2 \rightarrow X_3$ are identifiable.*

- ***Type (f3):*** *if $s_1 s_2 s_3^3 \notin \mathcal{M}(\mathcal{G})$ holds, then the target edges $X_1 \circ\!\rightarrow X_2$ and $X_2 \circ\!\rightarrow X_3$ are identifiable, provided that the two latent variables are distinct.*

- ***Type (f4):*** *if $s_1 s_2^2 s_3^3 \notin \mathcal{M}(\mathcal{G})$ holds, then the target edges $X_1 \circ\!\rightarrow X_2$ and $X_2 \circ\!\rightarrow X_3$ are identifiable.*

Collectively, the theorems above cover the entire search space for 3-node subgraphs. We conclude this section by formally stating the completeness of identifiability results.

**Theorem 3.13** (Completeness of the identifiability). *The identifiability conditions presented in Theorems 3.7 to 3.12 are complete with respect to monomial equivalence.*

To establish this validity, we systematically explore the structural space of three observed variables. After filtering for graph isomorphism, we identify 29 unique monomial equivalence classes. We verify that the proposed identifiability conditions are sufficient to uniquely recover the underlying structure for every case in this complete set. That is, for any 3-node subgraph in an LC-PB-SCM, the underlying PPADMG structure is identifiable if and only if the distinct

monomial sets satisfy Theorems 3.7 to 3.12. All proofs are given in the Appendix B.

Note that Theorem 3.13 provides a theoretical guarantee that every 3-node subgraph is maximally identifiable. For the global structure, it is possible to further identify the causal structure by exploiting additional information, e.g., structural priors or PGFs involving more than three variables. Extending identifiability to larger local substructures is a promising direction for future work. Nonetheless, the 3-node identifiability is general enough that for larger structures, one can enumerate these local structures to reconstruct the overall causal graph. This motivates the practical algorithm in the following section.

## 4. Practical algorithm

We propose a stepwise algorithm (Alg. 1) to learn the PPADMG of an LC-PB-SCM[1]. Given the observed count data $\mathcal{D}$, we utilize the empirical Local-PGF, denoted as $\hat{G}_{\mathbf{X}}^{\mathbf{L}}(\mathbf{s})$, based on Definition 2.5 and the empirical PGF formulation in (Nakamura & Pérez-Abreu, 1993). In particular, to test whether the partial derivative of the log-PGF is zero, one may employ a threshold or use the bootstrap hypothesis test (Efron & Tibshirani, 1994).

**Phase 1: Skeleton Learning.** We determine the adjacency between any pair $X_i, X_j$ by testing the existence of the monomial $s_i s_j$ is non-zero. In practice, we compute the mixed second-order partial derivative $\frac{\partial^2 \log \hat{G}_{\mathbf{X}}^{\{i,j\}}(\mathbf{s})}{\partial s_i \partial s_j}$ and test whether it is zero.

**Phase 2: Latent Confounding Detection.** For adjacent pairs, we identify latent confounders (oriented as $X_i \rightarrow X_j$ in the presence of $L \rightarrow \{X_i, X_j\}$) by checking for the monomial $s_i s_j^2$ (Theorem 3.8). This corresponds to testing the third-order derivative $\frac{\partial^3 \log \hat{G}_{\mathbf{X}}^{\{i,j\}}(\mathbf{s})}{\partial s_i \partial s_j^2}$ is zero.

**Phase 3: Orientation via Local Structures.** Finally, we orient the remaining edges within 3-node substructures by verifying the monomials as derived in Theorems 3.9 to 3.12.

## 5. Experiments

### 5.1. Synthetic Experiments

In this section, we evaluate our method on synthetic data generated from LC-PB-SCMs. We present the case study involving three observed variables to demonstrate the effectiveness of our method in recovering the causal structure of LC-PB-SCM from observed count data. The baseline methods includes FCI (Spirtes et al., 2000), GES (Chickering, 2002), OCD (Ni & Mallick, 2022), and PB-SCM (Xiang

---

[1]The code is available at https://github.com/DMIRLAB-Group/LC-PB-SCM.

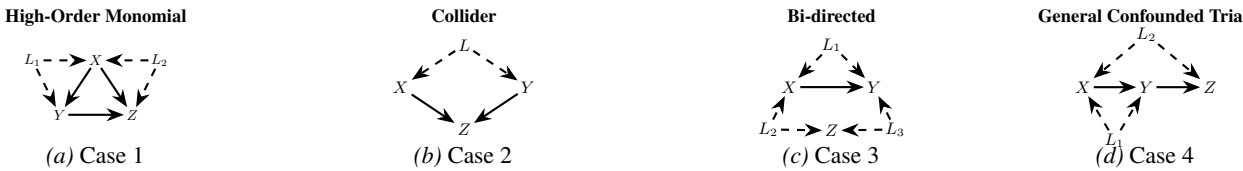

*Figure 4.* Overview of 4 causal structures. Nodes are observed variables $(X, Y, Z)$ and latent variables $(L)$.

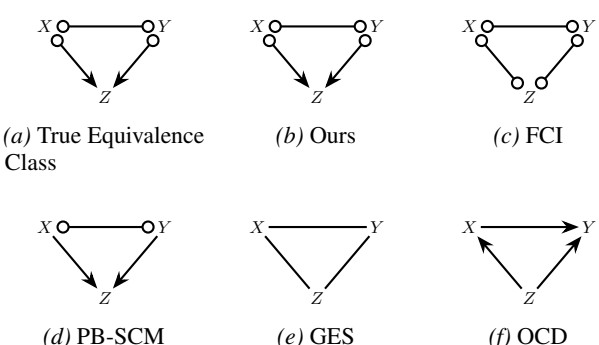

*Figure 5.* Comparison of the ground-truth PPADMG and the graphs recovered by different causal discovery methods in Case 2.

*Table 1.* Performance comparison measured by F1 scores.

| Method | Case 1 | Case 2 | Case 3 | Case 4 |
|--------|--------|--------|--------|--------|
| **Ours** | **1.00±0.00** | **0.72±0.19** | **0.84±0.05** | **0.87±0.15** |
| FCI | 0.00±0.00 | 0.53±0.00 | 0.00±0.00 | 0.00±0.00 |
| PB-SCM | 0.47±0.00 | 0.42±0.00 | 0.47±0.12 | 0.41±0.07 |
| GES | 0.55±0.00 | 0.33±0.00 | 0.37±0.00 | 0.44 ±0.02 |
| OCD | 0.46±0.11 | 0.33±0.03 | 0.66±0.08 | 0.58±0.13 |

et al., 2024). The default settings are as follows, sample size = 10000, range of causal coefficients $\alpha_{i,j} \in [0.1, 0.9]$, range of the mean of Poisson noise $\mu_i \in [0.02, 0.08]$. Each simulation is repeated 50 times. Performance is assessed by converting the estimated graphs into PPADMG and computing the F1-score against the true PPADMG derived from the ground-truth DAG. Given space constraints, key results are summarized in Table 1, which evaluates our method across varying typical latent confounding scenarios to illustrate its advantage over baselines under latent confounding. We further visualize a representative result for Case 2 in Fig. 5. The ground-truth PPADMG is $X \circ\!\!-\!\!\circ Y$ and $X \circ\!\!\rightarrow Z \leftarrow\!\!\circ Y$, which is correctly recovered by our method. In contrast, FCI only recovers the skeleton, PB-SCM misses the latent-confounding-induced uncertainty, GES returns an undirected triangle, and OCD produces incorrect edge orientations. Moreover, we have conducted additional experiments that cover all proposed theorems, along with tests on more than three variables to further assess algorithm performance (See Appendix C.1 for details).

## 5.2. Real-world Experiments

In this section, we evaluate the performance of our proposed method on real-world datasets to assess its effectiveness in real-world scenarios. We conduct our method on a *shopping mall paid search campaign dataset* collected from a U.S. shopping mall between July and November 2021[2]. The study involves the marketing funnel variables *Impressions*, *Clicks*, and *Conversions*. As shown in Figure 6, *Impressions* acts as a natural common cause for both *Clicks* and *Conversions*. By modeling *Impressions* as a latent confounder, we aim to verify if the method can successfully identify the causal direction from *Clicks* to *Conversions* despite the presence of confounding. Empirically, our method accurately recovered the causal direction *Clicks* → *Conversions*. This finding aligns with our theoretical results and corroborates the method's practical utility in analyzing real-world count data. We further evaluate our method using the *Football Events Dataset*, with results detailed in Appendix C.2.

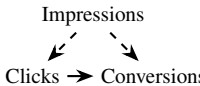

*Figure 6.* Causal Graph of Marketing Funnel

## 6. Conclusion

In this paper, we tackle the challenge of causal identification in PB-SCM under the presence of latent confounding. By introducing the Trie-based graphical representation, we establish a rigorous correspondence between the topological structure of causal paths and the algebraic monomials in the logarithmic probability generating function. This theoretical bridge allows us to define the Monomial Equivalence Class and formally establishes the identifiability condition of the LC-PB-SCM graphically based on the monomial. We further prove the completeness of identifiability rules for local substructures, showing that the proposed conditions are complete. Building on these theoretical insights, we develop a practical algorithm that leverages higher-order derivatives of the empirical PGF for causal discovery in LC-PB-SCM. Experiments on synthetic data and real-world applications demonstrate the performance of our approach, providing a robust solution for causal discovery in discrete count data where traditional methods often fail.

---

[2]https://www.kaggle.com/datasets/marceaxl82/shopping-mall-paid-search-campaign-dataset

## Acknowledgments

This research was supported in part by National Key R&D Program of China (2021ZD0111501), National Science Fund for Excellent Young Scholars (62122022), Natural Science Foundation of China (62406080, 61876043, 61976052, 62476163, 6244100778, U24A20233), the major key project of PCL (PCL2021A12), the National Key Research and Development Program of China (2025YFC3410000, 2025YFC3410003), the Natural Science Foundation of Guangdong Province (23201910250000514). ZM would like to acknowledge the support by STU Scientific Research Initiation Grant under grant No. NTF25024T. We sincerely appreciate the comments from anonymous reviewers, which greatly helped to improve the paper.

## Impact Statement

This paper presents work whose goal is to advance the field of machine learning, specifically in the domain of causal discovery from discrete count data. Count data is ubiquitous in high-stakes fields such as epidemiology (e.g., disease propagation), economics, and social sciences. By enabling the identification of causal structures in the presence of latent confounders, our method contributes to more reliable data analysis and can help mitigate the risk of deriving spurious correlations in observational studies. This has the potential to support better decision-making in public policy and business strategies. However, as with any causal discovery algorithm, there is a risk of misuse if the model's assumptions (e.g., the specific functional form of PB-SCM) are violated or if the output is interpreted without sufficient domain expertise. Incorrect causal conclusions in sensitive domains could lead to ineffective or harmful interventions. We encourage practitioners to validate the learned structures before deploying them in critical applications.

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

# Appendix

## A. Closed-Form Solution for PGF of LC-PB-SCM

We begin with a closed-form solution for PGF of PB-SCM, which is developed in (Xiang et al., 2024).

**Theorem A.1** (Closed-form solution for PGF of PB-SCM (Xiang et al., 2024)). *Given a random vector* $\mathbf{X} = [X_1, ..., X_n]^T$ *following PB-SCM, let* $\mathbf{s}_{(j)} = \{s_l | l \in Des(j) \cup \{j\}\}$, *the PGF of* $P(\mathbf{X})$ *is given by*

$$G_{\mathbf{X}}(\mathbf{s}) = \prod_{i \in [d]} G_{\epsilon_i}\left( s_i \times \prod_{j \in Ch(i)} G_{i,j}(\mathbf{s}_{(j)}) \right), \tag{A.1}$$

*where*

$$G_{i,j}(\mathbf{s}_{(j)}) = \begin{cases} G_{B(\alpha_{i,j})}\left( s_j \times \prod_{k \in Ch(j)} G_{j,k}(\mathbf{s}_{(k)}) \right) & , Ch(j) \neq \varnothing \\ G_{B(\alpha_{i,j})}(s_j) & , Otherwise \end{cases}, \tag{A.2}$$

*in which* $G_{\epsilon_i}(\cdot)$ *is the PGF of Poisson noise* $\epsilon_i$ *and* $G_{B(\alpha_{i,j})}(\cdot)$ *is the PGF of Bernoulli distribution with parameter* $\alpha_{i,j}$.

Based on Theorem A.1, we aim to develop a closed-form solution for PGF of LC-PB-SCM and further provide a simplified resolution for developing the Trie-based representation theorem.

**Theorem A.2** (Closed-Form Solution for PGF of LC-PB-SCM). *Consider the LC-PB-SCM as defined. For any node $i \in \mathbf{V}$, define an argument function $\phi_i(\mathbf{s})$ recursively as:*

$$\phi_i(\mathbf{s}) = s_i \prod_{k \in Ch(i)} \left( \alpha_{i,k} \phi_k(\mathbf{s}) + 1 - \alpha_{i,k} \right), \tag{A.3}$$

*where $\phi_i(\mathbf{s}) = s_i$, if $Ch(i) = \varnothing$, and $\mathbf{s}$ is the vector of PGF arguments. The joint PGF of all variables $\mathbf{X}$ is given by:*

$$G_{\mathbf{X}}(\mathbf{s}) = \exp\left( \sum_{i \in \mathbf{V}} \mu_i(\phi_i(\mathbf{s}) - 1) \right). \tag{A.4}$$

*Proof.* The LC-PB-SCM, defined over the union of observed and latent variables $\mathbf{X} = \mathbf{X_O} \cup \mathbf{X_L}$, structurally satisfies the definition of the generic PB-SCM addressed in Theorem A.1. Thus, we can directly apply Theorem A.1 by specifying the functional forms of the noise and edge distributions.

First, recall that in the LC-PB-SCM, we have exogenous noise (and root latent variables) follows a Poisson distribution whose PGF is $G_{\epsilon_i}(s) = \exp(\mu_i(s-1))$. The causal mechanism is Binomial thinning, which corresponds to a Bernoulli operation on the edges, and the PGF is $G_{B(\alpha_{i,j})}(s) = \alpha_{i,j}s + (1 - \alpha_{i,j})$.

Let $\Psi_i(\mathbf{s})$ be the argument function passed to the noise PGF $G_{\epsilon_i}$ in Theorem A.1. According to Eq. A.1, this argument is defined as:

$$\Psi_i(\mathbf{s}) = s_i \times \prod_{j \in Ch(i)} G_{i,j}(\mathbf{s}_{(j)}). \tag{A.5}$$

Next, substituting the Bernoulli PGF form $G_{B(\alpha_{i,j})}(s) = \alpha_{i,j}s + 1 - \alpha_{i,j}$ in Eq. A.2, the term $G_{i,j}(\mathbf{s}_{(j)})$ becomes:

$$G_{i,j}(\mathbf{s}(j)) = \begin{cases} \alpha_{i,j} \left( s_j \times \prod_{k \in Ch(j)} G_{j,k}(\mathbf{s}(k)) \right) + 1 - \alpha_{i,j}, & \text{if } Ch(j) \neq \varnothing \\ \alpha_{i,j} s_j + 1 - \alpha_{i,j}, & \text{otherwise.} \end{cases} \tag{A.6}$$

Observe the term inside the parentheses in the first case. It is structurally identical to the definition of $\Psi_j(\mathbf{s})$. Similarly, for the leaf case, $s_j$ is simply $\Psi_j(\mathbf{s})$. Thus, by combining Eq. A.5 and Eq. A.5, we can define the recursive function $\phi_j(\mathbf{s}) \equiv \Psi_j(\mathbf{s})$ which unifies both cases:

$$\Psi_i(\mathbf{s}) = s_i \prod_{j \in Ch(i)} \left( \alpha_{i,j} \Psi_j(\mathbf{s}) + 1 - \alpha_{i,j} \right). \tag{A.7}$$

This recursion is identical to the definition of $\phi_i(\mathbf{s})$ in Eq. (A.3). Thus, $\Psi_i(\mathbf{s}) \equiv \phi_i(\mathbf{s})$.

Finally, Theorem A.1 states that the joint PGF is the product of the noise PGFs evaluated at these effective arguments:

$$\begin{aligned} G_{\mathbf{X}}(\mathbf{s}) &= \prod_{i \in \mathbf{V}} G_{\epsilon_i}(\Psi_i(\mathbf{s})) \\ &= \prod_{i \in \mathbf{V}} \exp\left( \mu_i(\Psi_i(\mathbf{s}) - 1) \right). \end{aligned} \tag{A.8}$$

Substituting $\Psi_i(\mathbf{s}) = \phi_i(\mathbf{s})$ and summing the exponents:

$$G_{\mathbf{X}}(\mathbf{s}) = \exp\left( \sum_{i \in \mathbf{V}} \mu_i(\phi_i(\mathbf{s}) - 1) \right). \tag{A.9}$$

This completes the proof. $\qquad\square$

# B. Proof of Identifiability

### B.1. Proof of Theorem 3.5

We begin with the child subtree definition in order to characterize the recursive structure in Trie.

**Definition B.1** (Child Subtree). Let $\mathcal{T} = (\mathbf{N}', \mathbf{E}')$ and $\mathcal{T} \in \mathbb{S}(\mathcal{T}_\mathcal{G}(v_i))$ be a Pruned Trie or Trie that rooted at $i$. For a child vertex $j \in Ch_\mathcal{T}(i)$ of tree $\mathcal{T}$, we define the subtree rooted at $j$ in $\mathcal{T}$ as a child subtree, denoted by $\mathcal{T}_j$. Here, $\mathcal{T}_j$ consists of all nodes and edges in $\mathcal{T}$ that are descendants of $j$.

We now provide the proof of Theorem 3.5 as follows.

*Proof.* We prove by induction. Let $\Phi_i(\mathbf{s})$ denote the sum over the space of pruned tries rooted at $v_i$:

$$\Phi_i(\mathbf{s}) := \sum_{\mathcal{T}^* \in \mathbb{S}(\mathcal{T}_\mathcal{G}(v_i))} c(\mathcal{T}^*)m(\mathcal{T}^*) \tag{B.1}$$

We aim to show that $\Phi_i(\mathbf{s})$ satisfies the same recurrence relation as $\phi_i(\mathbf{s})$ in Eq. (A.3).

**Base Case:** Consider a leaf node $i$ in $\mathcal{G}$ (i.e., $Ch_\mathcal{G}(i) = \varnothing$). The space $\mathbb{S}(\mathcal{T}_\mathcal{G}(v_i))$ contains only a single tree consisting of the root node $i$ itself. For this tree, $\mathbf{E}^* = \varnothing$, $\mathcal{E}^*_{cut} = \varnothing$, and $\mathbf{N}^* = \{i\}$. Thus, $c(\mathcal{T}^*) = 1$ and $m(\mathcal{T}^*) = s_i$. We have $\Phi_i(\mathbf{s}) = s_i$, which matches the base case of $\phi_i(\mathbf{s})$.

**Inductive Step:** Consider a node $i$ with children set $Ch_\mathcal{G}(i)$. Assume that for all $j \in Ch_\mathcal{G}(i)$, $\Phi_j(\mathbf{s}) = \phi_j(\mathbf{s})$.

Any Pruned Trie $\mathcal{T}^* \in \mathbb{S}(\mathcal{T}_\mathcal{G}(v_i))$ is uniquely characterized by the root $i$ and the configuration of the edges connecting to its children. For each child $j \in Ch_\mathcal{G}(i)$, the edge $(i, j)$ is either:

1. **Pruned** (in $\mathcal{E}^*_{cut}$): Contributes factor $(1 - \alpha_{i,j})$ to $c(\mathcal{T}^*)$. No nodes from branch $j$ are included.

2. **Active** (in $\mathbf{E}^*$): Contributes factor $\alpha_{i,j}$ to $c(\mathcal{T}^*)$. The branch leads to a valid child pruned subtree $\mathcal{T}_j \in \mathbb{S}(\mathcal{T}_\mathcal{G}(v_j))$.

Let $Ch_{\mathcal{T}^*}(i)$ be the set of children connected by active edges (i.e., the edges does not pruned by $\mathcal{E}^*_{cut}$) in a specific Trie $\mathcal{T}^*$. The coefficient and monomial can be factorized recursively:

$$c(\mathcal{T}^*) = \left( \prod_{j \in Ch_{\mathcal{T}^*}(i)} \alpha_{i,j} c(\mathcal{T}_j) \right) \cdot \left( \prod_{k \in Ch_\mathcal{G}(i) \backslash Ch_{\mathcal{T}^*}(i)} (1 - \alpha_{i,k}) \right) \tag{B.2}$$

$$m(\mathcal{T}^*) = s_i \cdot \prod_{j \in Ch_{\mathcal{T}^*}(i)} m(\mathcal{T}_j) \tag{B.3}$$

Thus, based on Eq. B.2, we can rewrite Eq. B.1 as follows:

$$\begin{aligned}\Phi_i(\mathbf{s}) &= \sum_{\mathcal{T}^* \in \mathbb{S}(\mathcal{T}_\mathcal{G}(v_i))} c(\mathcal{T}^*)m(\mathcal{T}^*) \\ &= \sum_{\mathcal{T}^* \in \mathbb{S}(\mathcal{T}_\mathcal{G}(v_i))} \left( s_i \prod_{j \in Ch_{\mathcal{T}^*}(i)} \alpha_{i,j} c(\mathcal{T}_j)m(\mathcal{T}_j) \right) \cdot \left( \prod_{k \in Ch_\mathcal{G}(i) \backslash Ch_{\mathcal{T}^*}(i)} (1 - \alpha_{i,k}) \right)\end{aligned} \tag{B.4}$$

Since the set of all tree in $\mathbb{S}(\mathcal{T}_\mathcal{G}(v_i))$ can be enumerated by iterating over all possible subsets $S \subseteq Ch_\mathcal{G}(i)$ of children (i.e., enumerating all possible child edges in $\mathcal{G}$ to decide each edge whether active or not), and for each active child $j \in S$, iterating over all possible sub-tries $\mathcal{T}_j \in \mathbb{S}(\mathcal{T}_\mathcal{G}(v_j))$. Thus, by expanding the each tree in set $\mathbb{S}(\mathcal{T}_\mathcal{G}(v_i))$, we have

$$\Phi_i(\mathbf{s}) = \sum_{S \subseteq Ch_\mathcal{G}(i)} \sum_{\substack{\mathcal{T}_j \in \mathbb{S}(\mathcal{T}_\mathcal{G}(v_j)) \\ \forall j \in S}} \left[ s_i \left( \prod_{j \in S} \alpha_{i,j} c(\mathcal{T}_j)m(\mathcal{T}_j) \right) \left( \prod_{k \in Ch_\mathcal{G}(i) \backslash S} (1 - \alpha_{i,k}) \right) \right]$$

We can pull out $s_i$ and the terms that do not depend on the inner summation. Crucially, the choices of sub-tries for different children $j$ are independent. This allows us to push the summation inside the product:

$$\Phi_i(\mathbf{s}) = s_i \sum_{S \subseteq Ch_{\mathcal{G}}(i)} \left[ \left( \prod_{k \in Ch_{\mathcal{G}}(i) \setminus S} (1 - \alpha_{i,k}) \right) \cdot \prod_{j \in S} \left( \alpha_{i,j} \sum_{\mathcal{T}_j \in \mathbb{S}(\mathcal{T}_{\mathcal{G}}(v_j))} c(\mathcal{T}_j) m(\mathcal{T}_j) \right) \right]$$

By the inductive hypothesis, the inner sum is exactly $\phi_j(\mathbf{s})$. Thus:

$$\Phi_i(\mathbf{s}) = s_i \sum_{S \subseteq Ch_{\mathcal{G}}(i)} \left[ \prod_{j \in S} \alpha_{i,j} \phi_j(\mathbf{s}) \cdot \prod_{k \in Ch_{\mathcal{G}}(i) \setminus S} (1 - \alpha_{i,k}) \right] \tag{B.5}$$

The expression above is the expansion of a product of sums (by the General Distributive Law). Specifically, for each $j \in Ch_{\mathcal{G}}(i)$, we effectively choose between the term $\alpha_{i,j} \phi_j(\mathbf{s})$ (when $j \in S$) and the term $(1 - \alpha_{i,j})$ (when $j \notin S$). Therefore, the summation over all subsets $S$ collapses into:

$$\Phi_i(\mathbf{s}) = s_i \prod_{j \in Ch_{\mathcal{G}}(i)} (\alpha_{i,j} \phi_j(\mathbf{s}) + (1 - \alpha_{i,j})) \tag{B.6}$$

This matches the recursive definition of $\phi_i(\mathbf{s})$. Finally, substituting this into the expression for $G_{\mathbf{X}}(\mathbf{s})$, we obtain:

$$\sum_{i \in \mathbf{V}} \mu_i(\Phi_i(\mathbf{s}) - 1) = \sum_{i \in \mathbf{V}} \mu_i(\phi_i(\mathbf{s}) - 1) = \log G_{\mathbf{X}}(\mathbf{s}) \tag{B.7}$$

This completes the proof. □

## B.2. Proof of Theorem 3.7

*Proof.* The proof establishes the equivalence between the adjacency in the PPADMG $\mathcal{P}$ and the existence of the monomial $s_1 s_2$ in the log-PGF.

1. Sufficiency ($\Rightarrow$): Suppose $X_1$ and $X_2$ are adjacent in $\mathcal{P}$. By the definition of PPADMG, this adjacency implies either a direct edge or a latent common cause in the underlying graph $\mathcal{G}$.

- Direct Edge: There exists a path $X_1 \to X_2$ (or $X_1 \leftarrow X_2$. Without loss of generality, we can construct a Pruned Trie $\mathcal{T}^* \in \mathbb{S}(\mathcal{T}_{\mathcal{G}}(X_1))$ by selecting this specific path and pruning all other branches. The node set is $\mathbf{N}_{\mathcal{T}}^* = \{X_1, X_2\}$ and we have $m(\mathcal{T}^*) = s_1 s_2$.

- Latent Confounder ($X_1 \leftrightarrow X_2$): There exists a latent root $L$ with paths $L \to X_1$ and $L \to X_2$. We construct $\mathcal{T}^* \in \mathbb{S}(\mathcal{T}_{\mathcal{G}}(L))$ by retaining only these two paths. The observed nodes are exactly $\{X_1, X_2\}$, yielding $m(\mathcal{T}^*) = s_1 s_2$.

In both cases, by Theorem 3.5 and the faithfulness assumption (generic parameters), the coefficient $c(\mathcal{T}^*)$ is non-zero, ensuring $s_1 s_2 \in \mathcal{M}(\mathcal{G})$.

2. Necessity ($\Leftarrow$): Suppose $s_1 s_2 \in \mathcal{M}(G)$. By Theorem 3.5, there exists a Pruned Trie $\mathcal{T}^*$ such that its observed node set is exactly $\{X_1, X_2\}$. Let $v_{root}$ be the root of $\mathcal{T}^*$.

- If $v_{root} \in \{X_1, X_2\}$ (say $X_1$), $\mathcal{T}^*$ contains a directed path from $X_1$ to $X_2$. Since no other observed variables are in $\mathcal{T}^*$, this path consists solely of latent nodes, inducing a direct edge $X_1 \to X_2$ in $\mathcal{P}$, and vice versa.

- If $v_{root} \in \mathbf{L}$, $\mathcal{T}^*$ contains paths from $v_{root}$ to $X_1$ and $X_2$. Since $\{X_1, X_2\}$ are the only observed nodes, these paths define a latent confounder structure, inducing a bidirected edge $X_1 \leftrightarrow X_2$ in $\mathcal{P}$.

Therefore, the existence of $s_1 s_2$ implies $X_1$ and $X_2$ are adjacent in $\mathcal{P}$. □

## B.3. Proof of Theorem 3.8

*Proof.* Similar to Theorem 3.7, the proof relies on the unique correspondence between the structural paths and Trie.

1. Sufficiency ($\Rightarrow$): Suppose the causal structure contains a triangle: a latent confounder $L$ with edges $L \to X_1$, $L \to X_2$, and a directed edge $X_1 \to X_2$. Consider the Trie $\mathcal{T}_{\mathcal{G}}(L)$ rooted at the latent confounder $L$. There exist two distinct paths from $L$ to $X_2$, including $\pi_1 = L \to X_2$ and $\pi_2 = L \to X_1 \to X_2$. We can construct a Pruned Trie $\mathcal{T}^* \in \mathbb{S}(\mathcal{T}_{\mathcal{G}}(L))$ by retaining exactly these two paths and pruning all other branches such that $m(\mathcal{T}^*) = s_1 s_2^2$.

2. Necessity ($\Leftarrow$): Suppose $s_1 s_2^2 \in \mathcal{M}(\mathcal{G})$. This implies there exists a Pruned Trie $\mathcal{T}^*$ containing exactly one node for $X_1$ and two distinct nodes for $X_2$. Let $v_{root}$ be the root of $\mathcal{T}^*$. First, we determine the root $v_{root}$.

- If $v_{root} = X_1$: The path to $X_1$ is the root itself. To generate $s_2^2$, there must be two distinct directed paths from $X_1$ to $X_2$ without passing any other observed variables. In a simple causal graph without parallel edges, this is impossible. Thus, the root cannot be $X_1$ (nor $X_2$ by acyclicity).

- Therefore, $v_{root}$ must be a latent variable $L$.

Next, we analyze the structure of paths from $L$. Since $X_1$ appears once, there is a unique path $\pi_A : L \to X_1$. Moreover, since $X_2$ appears twice, there are two distinct paths $\pi_B, \pi_C : L \to X_2$. To form a coherent structure where $X_1$ and $X_2$ are adjacent (as given) and obtain exactly this monomial: One path to $X_2$ (say $\pi_C$) must pass through $X_1$. If it does not, $X_1$ would essentially be disconnected from the $X_2$ generation process in this Trie, or require a separate independent cause, which contradicts the single-Trie origin of the monomial $s_1 s_2^2$. Consequently, the existence of $s_1 s_2^2$ uniquely identifies the triangle structure $X_1 \leftarrow L \to X_2$ combined with $X_1 \to X_2$. $\square$

## B.4. Proof of Theorem 3.9

*Proof.* The proof analyzes the topological requirements for generating specific monomials in the Trie-based representation. We denote $X \circ\!\!\to Y$ as the existence of a directed edge into $Y$ (either $X \to Y$ or via latent confounding $X \leftrightarrow Y$ where the arrowhead is at $Y$).

**Type (c1): Shielded Collider**   We are given that $X_1$ and $X_2$ are adjacent. The condition is $s_1 s_2 s_3^2 \in \mathcal{M}(\mathcal{G})$. We prove this implies $X_1 \circ\!\!\to X_3$ and $X_2 \circ\!\!\to X_3$.

Suppose that the direction $X_3 \to X_1$ exists. The monomial $s_1 s_2 s_3^2$ requires a Trie root at some node $R$ to contain one node for $X_1$ and one for $X_2$, and two distinct path-induced nodes for $X_3$ (implying two distinct paths from $R$ to $X_3$).

$X_3$ must not be $R$ due to $s_3^2$ require another path to $X_3$, which leads to a cycle and is contradictory to acyclicity. Similarly, $X_1$ must not be $R$ due to $X_3 \to X_1$.

if $R = X_2$, then any directed path from $X_2 \to X_3$ must pass through $X_1$ (since $X_1$ and $X_2$ are adjacent and $X_3 \to X_1$). This yields at most one path to $X_3$, but generating $s_3^2$ requires branching paths to $X_3$, which is structurally infeasible when $X_3$ is an ancestor of the adjacent pair.

Moreover, if $R = L$ where $L$ is a latent confounder. If $L$ is a common cause of $X_1$ and $X_2$, paths flow to $X_1$ and $X_2$. The path from $L \to X_1$ does not pass through $X_3$ ($X_3 \to X_1$ block it). However, the path from $L \to X_2$ yields at most one path to $X_3$. If $L$ is a common cause of $X_2$ and $X_3$, paths flow to $X_2$ and $X_3$. The path from $L \to X_2$ does not pass through $X_3$ (It relies on the direction $X_2 \to X_3$ that would be identified by theorem 3.7). $L$ can not be the common cause of $X_1$ and $X_3$ (The existence of $X_3 \to X_1$ will form a Confounded Edge structure, which contradicts $X_1$ and $X_3$ is uncertain). Therefore, $R$ can not be $L$.

Thus, we can not find such an $R$, which means the assumption $X_3 \to X_1$ is incorrect. We conclude $X_1 \circ\!\!\to X_3$. Analogously, we can not find such an $R$ given that $X_3 \to X_2$, thus $X_2 \circ\!\!\to X_3$ is correct.

**Type (c2): Unshielded Collider**   We are given that $X_1$ and $X_2$ are non-adjacent. The condition is $s_1 s_2 s_3 \notin \mathcal{M}(\mathcal{G})$.

We proceed by contradiction. Suppose the target structure is *not* a collider, meaning at least one edge is directed away from $X_3$ (e.g., $X_3 \to X_1$). If $X_3 \to X_1$, since $X_1$ and $X_2$ are non-adjacent, the connection must be mediated by $X_3$.

- **Scenario A (Chain $X_2 - X_3 \to X_1$):** Consider the Trie rooted at the start of the chain (e.g., $X_2$ or a latent cause of $X_2$). There exists a path $X_2 \to X_3 \to X_1$. This Trie contains nodes $\{X_1, X_2, X_3\}$, generating the monomial $s_1 s_2 s_3$.

- **Scenario B (Fork $X_1 \leftarrow X_3 \to X_2$):** Consider the Trie rooted at $X_3$. It contains paths $X_3 \to X_1$ and $X_3 \to X_2$. The node set is $\{X_1, X_2, X_3\}$, generating $s_1 s_2 s_3$.

In both non-collider scenarios, the monomial $s_1 s_2 s_3$ exists. This contradicts the given condition $s_1 s_2 s_3 \notin \mathcal{M}(\mathcal{G})$. Therefore, the assumption that it is not a collider (e.g., $X_3 \to X_1$) must be false. We conclude that $X_1 \circ\!\!\to X_3$. Consequently, we can further rule out the direction $X_3 \to X_2$ similar to the **Scenario A** and conclude $X_3 \circ\!\!\to X_2$.

$\square$

### B.5. Proof of Theorem 3.10

*Proof.* We proved the theorem type by type.

**Type (d1)**  We analyze the structure where the context includes a latent triangle over $X_1, X_2$ (implying $s_1 s_2^2$) and an edge $X_2 \to X_3$. The condition is $s_1 s_2^2 s_3^3 \in \mathcal{M}(\mathcal{G})$.

The sub-monomial $s_1 s_2^2$ implies a Trie rooted at a latent confounder $L$ with paths: 1. $L \to X_1$ 2. $L \to X_2$ 3. $L \to X_1 \to X_2$

To satisfy the condition of $s_3^3$, there must be exactly three distinct paths reaching $X_3$ in this specific Trie. Given the context $X_2 \to X_3$, we can extend the existing paths ending at $X_2$:

- Extension 1: $L \to X_2 \to X_3$ (provides 1st $s_3$).

- Extension 2: $L \to X_1 \to X_2 \to X_3$ (provides 2nd $s_3$).

We are missing exactly one occurrence of $s_3$. The only remaining node in the Trie that can extend to $X_3$ is $X_1$ (from the path $L \to X_1$). Therefore, there must exist a direct edge $X_1 \to X_3$ to form the path $L \to X_1 \to X_3$, providing the 3rd $s_3$, or there exists another edge from $L$ to $X_3$ such that $X_1$ and $X_3$ are not adjacent. This concludes the causal direction of $X_1 \circ\!\!\to X_3$.

**Type (d2)**  The structural context is the latent triangle over $X_1, X_2$ (generated by $L$). The relations between $\{X_1, X_2\}$ and $X_3$ are unknown. The condition is $s_1 s_2^2 s_3^3 \in \mathcal{M}(\mathcal{G})$.

Similar to Type (d1), the base structure provides three endpoints capable of extending to $X_3$: 1. The node $X_1$ (from $L \to X_1$). 2. The node $X_2$ (from $L \to X_2$). 3. The node $X_2$ (from $L \to X_1 \to X_2$).

To generate $s_3^3$, *all three* endpoints must have a directed edge to $X_3$.

- Connecting the $X_1$ node to $X_3$ requires the edge $X_1 \to X_3$.

- Connecting both $X_2$ nodes to $X_3$ requires the edge $X_2 \to X_3$.

If either edge were missing, the total count of $s_3$ would be less than 3. For instance, if $X_1 \not\to X_3$, we lose the path $L \to X_1 \to X_3$, reducing the monomial to $s_1 s_2^2 s_3^2$ (assuming $X_2 \to X_3$ exists). Therefore, both $X_1 \to X_3$ and $X_2 \to X_3$ are identifiable.

**Type (d3)**  The structural context involves a latent triangle over $X_1, X_3$ (paths $L \to X_1$ and $L \to X_3$, plus $X_1 \to X_3$), which naturally generates a base monomial $s_1 s_3^2$. The condition is $s_1 s_2 s_3^3 \in \mathcal{M}(\mathcal{G})$.

We analyze the difference between the base ($s_1 s_3^2$) and the target ($s_1 s_2 s_3^3$). We need to account for one additional $s_2$ and one additional $s_3$.

1. Identifying $X_1 \to X_2$: The variable $s_2$ appears in the monomial. Since the root is $L$ and the path $L \to X_1$ exists, $X_2$ must be a descendant of $L$. Given the acyclic constraints and the fact that $X_2$ acts as a mediator in this specific high-order structure (appearing once), the valid path is $L \to X_1 \to X_2$. This identifies $X_1 \to X_2$.

2. Identifying $X_2 \to X_3$: Now the Trie contains a path to $X_2$ ($L \to X_1 \to X_2$). We still need one more $s_3$ to reach the count of 3. (Current $s_3$ sources: $L \to X_3$ and $L \to X_1 \to X_3$). The only available node to extend is the newly established $X_2$. By adding the edge $X_2 \to X_3$, we form the path $L \to X_1 \to X_2 \to X_3$. This provides the 3rd $s_3$.

Thus, the monomial $s_1 s_2 s_3^3$ uniquely identifies the chain structure $X_1 \to X_2 \to X_3$ within this context. $\square$

## B.6. Proof of Theorem 3.11

*Proof.* The proof relies on examining the Trie structures rooted at the latent confounder $L$ of $\{X_1, X_2\}$. This common cause, combined with the edge $X_1 \to X_2$, generates the base monomial signature $s_1 s_2^2$ (corresponding to paths $L \to X_1$, $L \to X_2$, and $L \to X_1 \to X_2$). We analyze how the target edges extend these paths to form specific exponents for $X_3$.

- **Type (e1):** The structural context specifies that $X_1$ and $X_3$ are **non-adjacent**. The condition is $s_1 s_2^2 s_3^2 \in \mathcal{M}(\mathcal{G})$. This implies the existence of exactly two directed paths from the root $L$ to $X_3$. Since $X_1$ cannot link to $X_3$ directly, the paths must pass through $X_2$. The base structure contains two "endpoints" at $X_2$ ($L \to X_2$ and $L \to X_1 \to X_2$). By extending both endpoints to $X_3$ via the edge $X_2 \to X_3$, we obtain exactly two paths to $X_3$. Therefore, the monomial $s_1 s_2^2 s_3^2$ uniquely identifies the target edge $X_2 \to X_3$.

- **Type (e2):** Note that we do not utilize the information regarding whether $X_2$ and $X_3$ are adjacent, and thus the adjacency between $X_2$ and $X_3$ is **optional**. The condition is $s_1 s_2^2 s_3^2 \in \mathcal{M}(\mathcal{G})$. We analyze the potential orientation of the edge between $X_2$ and $X_3$. If the edge were $X_3 \to X_2$, the paths accumulating at $X_2$ could not extend to $X_3$. The only path to $X_3$ would be via $X_1$ ($L \to X_1 \to X_3$, yielding at most $s_3^1$). Conversely, we can rule out $X_3 \to X_2$ and the remain possible edges ($X_2 \leftrightarrow X_3$ and $X_2 \to X_3$) imply the arrow-head at $X_3$ (i.e., $X_2 \circ\!\!\to X_3$).

- **Type (e3):** Note that we do not utilize the information regarding whether $X_2$ and $X_3$ are adjacent, and thus the adjacency between $X_2$ and $X_3$ is **optional**. The condition is $s_1 s_2^2 s_3^2 \notin \mathcal{M}(\mathcal{G})$. If the edge were $X_2 \to X_3$, the paths $L \to X_1 \to X_2 \to X_3$ and $L \to X_2 \to X_3$ would construct a Trie generating the monomial $s_1 s_2^2 s_3^2$ in log PGF. This contradicts to $s_1 s_2^2 s_3^2 \notin \mathcal{M}(\mathcal{G})$. Therefore, we can rule out $X_2 \to X_3$ and conclude $X_3 \circ\!\!\to X_2$.

- **Type (e4):** The structural context specifies that $X_2$ and $X_3$ are **non-adjacent**. The condition is $s_1 s_2^2 s_3 \in \mathcal{M}(\mathcal{G})$. We need exactly one path to $X_3$. Since $X_2$ is not connected to $X_3$, the paths passing through $X_2$ are blocked. The only remaining node in the base structure capable of reaching $X_3$ is $X_1$. Extending the path $L \to X_1$ to $X_3$ requires the directed edge $X_1 \to X_3$. This single path ($L \to X_1 \to X_3$) generates exactly $s_3^1$. Thus, the target edge $X_1 \to X_3$ is identified.

- **Type (e5):** The condition is $s_1 s_2^2 s_3 \notin \mathcal{M}(\mathcal{G})$. Similar to the type (e3), if $X_1 \to X_3$ existed, the path $L \to X_1 \to X_3$ would be formed, generating at least the monomial $s_1 s_2^2 s_3$. The explicit absence of $s_1 s_2^2 s_3$ rules out the existence of the path $L \to X_1 \to X_3$. Consequently, the adjacency between $X_1$ and $X_3$ must be directed as $X_3 \to X_1$ or $X_3 \leftrightarrow X_1$. Thus, $X_3 \circ\!\!\to X_1$ is identified.

$\square$

## B.7. Proof of Theorem 3.12

*Proof.* We proved the theorem type by type. Specifically, we proceed by contradiction and eliminating potential directed orientations to identify the bi-directed structure.

**Type (f1) three bi-directed edges structure**

- **Identifying $X_1 \leftrightarrow X_2$:** Suppose that the edge $X_1 \to X_2$ exists. Given the adjacency $X_3 \circ\!\!-\!\!\circ X_1$, this orientation implies the existence of a structure $X_3 \circ\!\!-\!\!\circ X_1 \to X_2$. By Theorem 3.5, such a structure will generate the monomial $s_{X_1} s_{X_2} s_{X_3}$ in the log-PGF, which leads to a contradiction. Therefore, the direction $X_1 \to X_2$ is ruled out. Similarly, we can also reject the edge $X_1 \leftarrow X_2$ by structure $X_1 \leftarrow X_2 \circ\!\!-\!\!\circ X_3$. Since node $X_1$ and $X_2$ are adjacent, but both directed orientations ($X_1 \to X_2$ and $X_2 \to X_1$) lead to contradictions with the Monomial Representation Theorem, the only admissible structure is the bidirectional edge $X_1 \leftrightarrow X_2$.

- **Identifying $X_2 \leftrightarrow X_3$:** Next, we proceed to rule out $X_2 \to X_3$ by structure $X_1 \leftrightarrow X_2 \to X_3$ and $X_2 \leftarrow X_3$ by structure $X_2 \leftarrow X_3 \circ\!\!-\!\!\circ X_1$. Therefore, we conclude $X_2 \leftrightarrow X_3$.

- **Identifying $X_1 \leftrightarrow X_3$:** Similarly, we rule out $X_1 \to X_3$ by structure $X_2 \leftrightarrow X_1 \to X_3$ and $X_1 \leftarrow X_3$ by structure $X_1 \leftarrow X_3 \circ\!\!-\!\!\circ X_2$. Consequently, we conclude $X_1 \leftrightarrow X_3$.

**Type (f2) two bi-directed edges structure** Let $L$ denote the confounder between $X_1$ and $X_2$.

- **Establishing edge orientations ($X_3 \circ\!\!\to X_1$ and $X_3 \circ\!\!\to X_2$):** Suppose that the directed edge $X_2 \to X_3$ exists. Under this assumption, the paths $L \to X_1 \to X_2$ and $L \to X_2 \to X_3$ would facilitate the construction of a Trie generating the

monomial $s_1 s_2^2 s_3$, which leads to a contradiction. Consequently, we determine the orientation $X_3 \circ\!\!\rightarrow X_2$. Analogously, we rule out the edge $X_1 \rightarrow X_3$ by analyzing the paths $L \rightarrow X_1 \rightarrow X_2$, $L \rightarrow X_2$, and $L \rightarrow X_1 \rightarrow X_3$, thereby concluding $X_3 \circ\!\!\rightarrow X_1$.

- **Identifying bi-directed edges** ($X_3 \leftrightarrow X_1$ and $X_3 \leftrightarrow X_2$): Next, we proceed to exclude $X_3 \rightarrow X_1$. This is achieved by leveraging the path $X_3 \rightarrow X_1 \rightarrow X_2$ in conjunction with the established relation $X_3 \circ\!\!\rightarrow X_2$ and we can conclude $X_3 \leftrightarrow X_1$. Finally, regarding the connection between $X_2$ and $X_3$, we exclude $X_3 \rightarrow X_2$ based on the structure $X_2 \leftarrow X_3 \leftrightarrow X_1$ and path $X_3 \rightarrow X_2$, which confirms the bi-directed edge $X_3 \leftrightarrow X_2$.

**Type (f3, f4) single bi-directed edge structure**

- **Type (f3):** Suppose that the directed edge $X_1 \rightarrow X_3$ exists. This configuration corresponds to the identifiable case described in Theorem 3.10 (specifically Case d3). The failure to match this identification condition leads to the rejection of the $X_1 \rightarrow X_3$ hypothesis. Analogously, we apply similar reasoning to rule out the reverse direction $X_3 \rightarrow X_1$. With both directed options invalidated, the only remaining valid orientation is the bi-directed edge $X_1 \leftrightarrow X_3$.

- **Type (f4):** First, the direction $X_3 \rightarrow X_1$ is explicitly ruled out by the constraint of acyclic. Next, we consider the alternative direction $X_1 \rightarrow X_3$. Had this directed edge existed, the structure would have been identified under the conditions of Theorem 3.10 (specifically Case d2). The fact that this structure was not detected by the theorem implies that the edge is not directed as $X_1 \rightarrow X_3$. Consequently, having excluded both directed possibilities, we conclude the existence of the bi-directed edge $X_3 \leftrightarrow X_1$.

$\square$

## B.8. Proof of Theorem 3.13

*Figure 7.* The complete skeleton of three observed variables up to permutations of variables.

*Proof.* To establish the validity of Theorem 3.13, we systematically explore the structural space of any three observed variables. The verification process is hierarchical: we first enumerate all potential underlying skeletons as shown in Figure 7. Then, conditioned on each skeleton, we exhaustively examine all valid permutations of edge directions and variable orderings. After accounting for isomorphism (where PGFs are identical upon variable relabeling), we find that there exist only 29 unique valid PGFs generated by three observed variables.

Specifically, we enumerate all possible Directed Acyclic Graphs (DAGs) over the three observed variables and any latent variables. For each edge endpoint that could have multiple possible directions, we mark it with a circle; for endpoints with a uniquely determined direction, we mark it with an arrowhead. This yields the corresponding ground truth PPADMGs, which are shown in Figure 8. We further establish that our proposed identifiability rules are sufficient to correctly recover the underlying structure for every case in this complete set. In other words, every arrow-head in Figure 8 could be identified by Theorem 3.7 to 3.11.

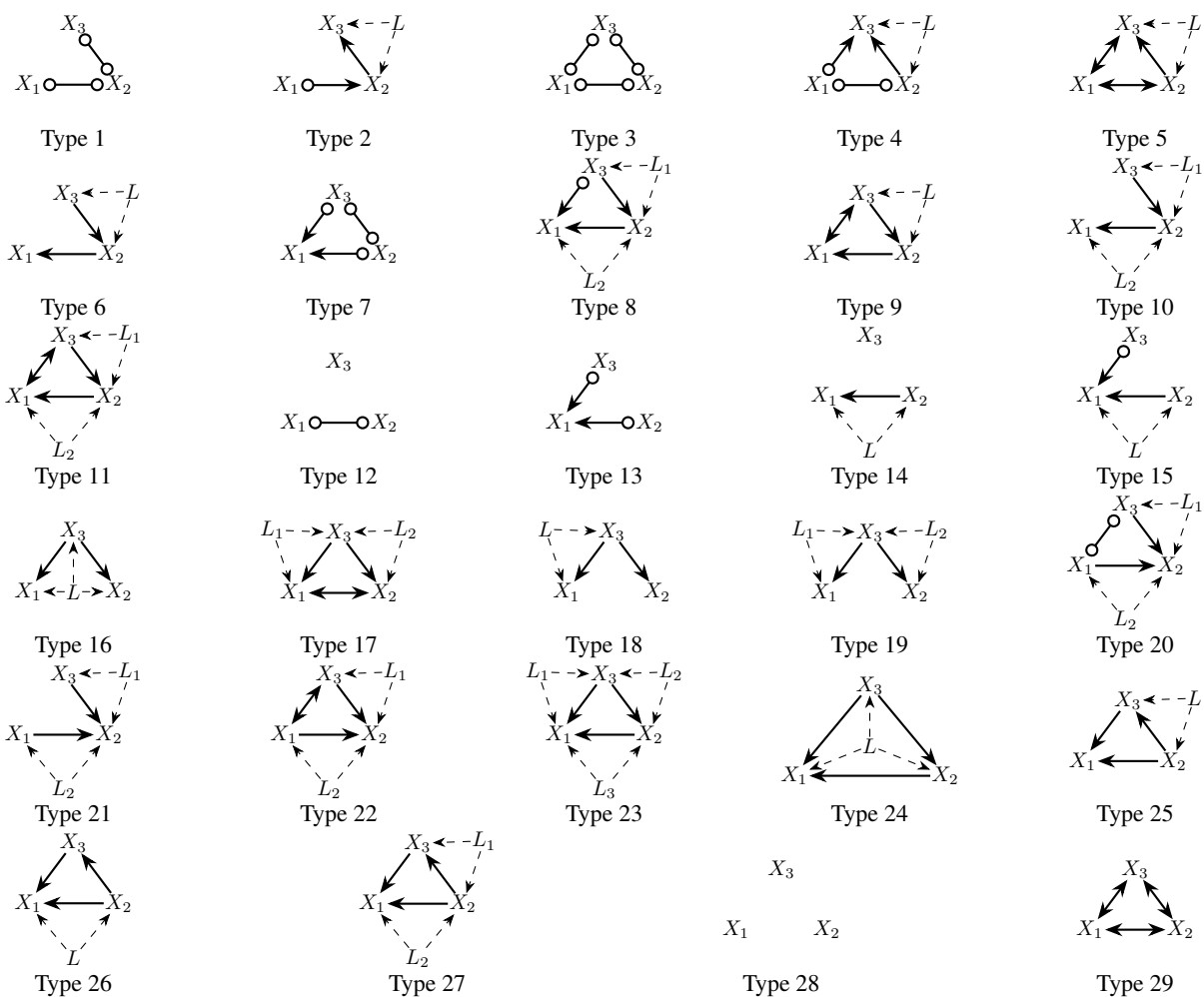

*Figure 8.* Overall 29 monomial equivalence classes in local 3-variables LC-PB-SCM.

Below, we demonstrate this verification for each equivalence class.

**Type 1** For all DAGs falling into this equivalence class, the set of distinct monomials is given by:

$$\mathcal{M}(\mathcal{G}) = \{s_1 s_2 s_3, s_1 s_2, s_1, s_2 s_3, s_2, s_3\}$$

Since there are no direct edges in this equivalence class, this means that every circle has two possible directions during the enumeration of the DAG under this equivalence class. Moreover, the monomial set $\mathcal{M}(\mathcal{G})$ does not exhibit any of the specific structural patterns required by Theorems 3.7 to 3.10 and 3.12. Consequently, our algorithm correctly identifies the equivalence class of Type 1.

**Type 2** For all DAGs falling into this equivalence class, the set of distinct monomials is given by:

$$\mathcal{M}(\mathcal{G}) = \{s_1 s_2 s_3, s_1 s_2, s_1, s_2 s_3^2, s_2 s_3, s_2, s_3\}$$

The monomial $s_2 s_3^2 \in \mathcal{M}(\mathcal{G})$ uniquely identifies the confounded directed edge $X_2 \to X_3$ via Theorem 3.8. For the edge between $X_1$ and $X_2$, the monomial set satisfies Type (e5) in Theorem 3.11, resulting in the circle-directed edge $X_1 \circ\!\!\to X_2$ in the equivalence class. Consequently, our algorithm correctly identifies the equivalence class of Type 2.

**Type 3**    For all DAGs falling into this equivalence class, the set of distinct monomials is given by:

$$\mathcal{M}(\mathcal{G}) = \{s_1 s_2 s_3, s_1 s_2, s_1 s_3, s_1, s_2 s_3, s_2, s_3\}$$

The set contains only second-order monomials representing adjacencies for all pairs $(s_1 s_2, s_1 s_3, s_2 s_3)$ and the third-order interaction. It lacks the specific higher-order patterns (e.g., $s_i s_j^2$) required to orient any edges as directed. Thus, all edges remain unoriented, represented as circle-circle edges ($X_1 \circ\!\!-\!\!\circ X_2$, $X_1 \circ\!\!-\!\!\circ X_3$, $X_2 \circ\!\!-\!\!\circ X_3$). Consequently, our algorithm correctly identifies the equivalence class of Type 3.

**Type 4**    For all DAGs falling into this equivalence class, the set of distinct monomials is given by:

$$\mathcal{M}(\mathcal{G}) = \{s_1 s_2 s_3^2, s_1 s_2 s_3, s_1 s_2, s_1 s_3, s_1, s_2 s_3^2, s_2 s_3, s_2, s_3\}$$

The monomial $s_2 s_3^2$ identifies the confounded directed edge $X_2 \rightarrow X_3$ via Theorem 3.8. Additionally, the absence of monomial $s_1^2 s_2 s_3^2$ can further identify the circle-directed edge $X_1 \circ\!\!\rightarrow X_3$ by theorem 3.11(e3). The edge $X_1 \circ\!\!-\!\!\circ X_2$ remains unoriented. Consequently, our algorithm correctly identifies the equivalence class of Type 4.

**Type 5**    For all DAGs falling into this equivalence class, the set of distinct monomials is given by:

$$\mathcal{M}(\mathcal{G}) = \{s_1 s_2 s_3, s_1 s_2, s_1 s_3, s_1, s_2 s_3^2, s_2 s_3, s_2, s_3\}$$

The presence of $s_2 s_3^2$ identifies the confounded directed edge $X_2 \rightarrow X_3$ via Theorem 3.8. The remaining bi-directed edges satisfy the Type (f2) in Theorem 3.12, leaving the edges $X_1 \leftrightarrow X_2$ and $X_1 \leftrightarrow X_3$ as bi-directed. Consequently, our algorithm correctly identifies the equivalence class of Type 5.

**Type 6**    For all DAGs falling into this equivalence class, the set of distinct monomials is given by:

$$\mathcal{M}(\mathcal{G}) = \{s_1^2 s_2^2 s_3, s_1 s_2^2 s_3, s_1 s_2 s_3, s_1 s_2, s_1, s_2^2 s_3, s_2 s_3, s_2, s_3\}$$

The monomial $s_2^2 s_3$ identifies the confounded directed edge $X_3 \rightarrow X_2$ via Theorem 3.8. Furthermore, the monomial $s_1^2 s_2^2 s_3$ corresponds to the specific chain structure identified in Type (e1) in Theorem 3.11, allowing for the identification of the directed edge $X_2 \rightarrow X_1$. Consequently, our algorithm correctly identifies the equivalence class of Type 6.

**Type 7**    For all DAGs falling into this equivalence class, the set of distinct monomials is given by:

$$\mathcal{M}(\mathcal{G}) = \{s_1^2 s_2 s_3, s_1 s_2 s_3, s_1 s_2, s_1 s_3, s_1, s_2 s_3, s_2, s_3\}$$

The monomial $s_1^2 s_2 s_3$ signifies a collider structure at $X_1$ according to Theorem 3.9. This pattern identifies the arrowheads at $X_1$, resulting in circle-directed edges $X_2 \circ\!\!\rightarrow X_1$ and $X_3 \circ\!\!\rightarrow X_1$. The edge between $X_2$ and $X_3$ remains a circle-circle edge $X_2 \circ\!\!-\!\!\circ X_3$. Consequently, our algorithm correctly identifies the equivalence class of Type 7.

**Type 8**    For all DAGs falling into this equivalence class, the set of distinct monomials is given by:

$$\mathcal{M}(\mathcal{G}) = \{s_1^3 s_2^2 s_3, s_1^2 s_2^2 s_3, s_1^2 s_2 s_3, s_1^2 s_2, s_1 s_2^2 s_3, s_1 s_2 s_3, s_1 s_2, s_1 s_3, s_1, s_2^2 s_3, s_2 s_3, s_2, s_3\}$$

This set identifies two confounded directed edges: $X_3 \rightarrow X_2$ (via $s_2^2 s_3$) and $X_2 \rightarrow X_1$ (via $s_1^2 s_2$) using Theorem 3.8. Additionally, the high-order monomial $s_1^3 s_2^2 s_3$ helps resolve the relation between $X_3$ and $X_1$, resulting in the circle-directed edge $X_3 \circ\!\!\rightarrow X_1$ by theorem 3.10(d1). Consequently, our algorithm correctly identifies the equivalence class of Type 8.

**Type 9**   For all DAGs falling into this equivalence class, the set of distinct monomials is given by:

$$\mathcal{M}(\mathcal{G}) = \{s_1^2 s_2^2 s_3, s_1^2 s_2 s_3, s_1 s_2^2 s_3, s_1 s_2 s_3, s_1 s_2, s_1 s_3, s_1, s_2^2 s_3, s_2 s_3, s_2, s_3\}$$

The monomial $s_2^2 s_3$ identifies the confounded directed edge $X_3 \to X_2$ via Theorem 3.8. The directed edge $X_2 \to X_1$ is identified via Theorem 3.11. The edge between $X_1$ and $X_3$ remains bi-directed as indicated by Theorem 3.12 patterns. Consequently, our algorithm correctly identifies the equivalence class of Type 9.

**Type 10**   For all DAGs falling into this equivalence class, the set of distinct monomials is given by:

$$\mathcal{M}(\mathcal{G}) = \{s_1^2 s_2^2 s_3, s_1^2 s_2, s_1 s_2^2 s_3, s_1 s_2 s_3, s_1 s_2, s_1, s_2^2 s_3, s_2 s_3, s_2, s_3\}$$

The monomials $s_2^2 s_3$ and $s_1^2 s_2$ uniquely identify the confounded directed edges $X_3 \to X_2$ and $X_2 \to X_1$, respectively, via Theorem 3.8. Consequently, our algorithm correctly identifies the equivalence class of Type 10.

**Type 11**   For all DAGs falling into this equivalence class, the set of distinct monomials is given by:

$$\mathcal{M}(\mathcal{G}) = \{s_1^2 s_2^2 s_3, s_1^2 s_2 s_3, s_1^2 s_2, s_1 s_2^2 s_3, s_1 s_2 s_3, s_1 s_2, s_1 s_3, s_1, s_2^2 s_3, s_2 s_3, s_2, s_3\}$$

The set identifies two confounded directed edges: $X_3 \to X_2$ (via $s_2^2 s_3$) and $X_2 \to X_1$ (via $s_1^2 s_2$) per Theorem 3.8. The remaining edge $X_1 \leftrightarrow X_3$ is identified as bi-directed by Type (f3) in Theorem 3.12. Consequently, our algorithm correctly identifies the equivalence class of Type 11.

**Type 12**   For all DAGs falling into this equivalence class, the set of distinct monomials is given by:

$$\mathcal{M}(\mathcal{G}) = \{s_1 s_2, s_1, s_2, s_3\}$$

This set contains only the adjacency $s_1 s_2$. Without any higher-order terms to orient the edge, it remains a circle-circle edge $X_1 \circ\!\!-\!\!\circ X_2$. Consequently, our algorithm correctly identifies the equivalence class of Type 12.

**Type 13**   For all DAGs falling into this equivalence class, the set of distinct monomials is given by:

$$\mathcal{M}(\mathcal{G}) = \{s_1 s_2, s_1 s_3, s_1, s_2, s_3\}$$

The set identifies adjacencies $s_1 s_2$ and $s_1 s_3$. It describes a V-structure centered at $X_1$, due to the lack of the explicit collider monomial $s_1 s_2 s_3$ by 3.9, resulting in circle-directed edges $X_2 \circ\!\!\rightarrow X_1$ and $X_3 \circ\!\!\rightarrow X_1$ in this equivalence class. Consequently, our algorithm correctly identifies the equivalence class of Type 13.

**Type 14**   For all DAGs falling into this equivalence class, the set of distinct monomials is given by:

$$\mathcal{M}(\mathcal{G}) = \{s_1^2 s_2, s_1 s_2, s_1, s_2, s_3\}$$

The monomial $s_1^2 s_2$ identifies the confounded directed edge $X_2 \to X_1$ via Theorem 3.8. Consequently, our algorithm correctly identifies the equivalence class of Type 14.

**Type 15**   For all DAGs falling into this equivalence class, the set of distinct monomials is given by:

$$\mathcal{M}(\mathcal{G}) = \{s_1^2 s_2, s_1 s_2, s_1 s_3, s_1, s_2, s_3\}$$

The monomial $s_1^2 s_2$ identifies the confounded directed edge $X_2 \to X_1$ via Theorem 3.8. The circle-directed edge $X_3 \circ\!\!\rightarrow X_1$ is identified by Type (e3) in Theorem 3.11. Consequently, our algorithm correctly identifies the equivalence class of Type 15.

**Type 16** For all DAGs falling into this equivalence class, the set of distinct monomials is given by:

$$\mathcal{M}(\mathcal{G}) = \{s_1^2 s_2^2 s_3, s_1^2 s_2 s_3, s_1^2 s_3, s_1 s_2^2 s_3, s_1 s_2 s_3, s_1 s_2, s_1 s_3, s_1, s_2^2 s_3, s_2 s_3, s_2, s_3\}$$

The monomials $s_2^2 s_3$ and $s_1^2 s_3$ identify the confounded directed edges $X_3 \rightarrow X_2$ and $X_3 \rightarrow X_1$, respectively, via Theorem 3.8. The bi-directed relationship $X_1 \leftrightarrow X_2$ is identified by Type (e2) Theorem 3.11 at both edge-marks (i.e., $X_1 \circ\!\!\rightarrow X_2$ and $X_1 \leftarrow\!\!\circ X_2$ by $s_1^2 s_2^2 s_3 \in \mathcal{M}(\mathcal{G})$). Consequently, our algorithm correctly identifies the equivalence class of Type 16.

**Type 17** For all DAGs falling into this equivalence class, the set of distinct monomials is given by:

$$\mathcal{M}(\mathcal{G}) = \{s_1^2 s_2 s_3, s_1^2 s_3, s_1 s_2^2 s_3, s_1 s_2 s_3, s_1 s_2, s_1 s_3, s_1, s_2^2 s_3, s_2 s_3, s_2, s_3\}$$

$s_2^2 s_3$ and $s_1^2 s_3$ identify the confounded directed edges $X_3 \rightarrow X_2$ and $X_3 \rightarrow X_1$ via Theorem 3.8. The bi-directed relationship $X_1 \leftrightarrow X_2$ is identified by Type (e3) Theorem 3.11 at both edge-marks (i.e., $X_1 \circ\!\!\rightarrow X_2$ and $X_1 \leftarrow\!\!\circ X_2$ by $s_1^2 s_2^2 s_3 \notin \mathcal{M}(\mathcal{G})$). Consequently, our algorithm correctly identifies the equivalence class of Type 17.

**Type 18** For all DAGs falling into this equivalence class, the set of distinct monomials is given by:

$$\mathcal{M}(\mathcal{G}) = \{s_1^2 s_2 s_3, s_1^2 s_3, s_1 s_2 s_3, s_1 s_3, s_1, s_2 s_3, s_2, s_3\}$$

The monomial $s_1^2 s_3$ identifies the confounded directed edge $X_3 \rightarrow X_1$ via Theorem 3.8. The adjacency $s_2 s_3$ is present and corresponds to the directed edge $X_3 \rightarrow X_2$ by Theorem 3.11. Consequently, our algorithm correctly identifies the equivalence class of Type 18.

**Type 19** For all DAGs falling into this equivalence class, the set of distinct monomials is given by:

$$\mathcal{M}(\mathcal{G}) = \{s_1^2 s_2 s_3, s_1^2 s_3, s_1 s_2^2 s_3, s_1 s_2 s_3, s_1 s_3, s_1, s_2^2 s_3, s_2 s_3, s_2, s_3\}$$

The monomials $s_1^2 s_3$ and $s_2^2 s_3$ uniquely identify the two confounded directed edges $X_3 \rightarrow X_1$ and $X_3 \rightarrow X_2$ via Theorem 3.8. Consequently, our algorithm correctly identifies the equivalence class of Type 19.

**Type 20** For all DAGs falling into this equivalence class, the set of distinct monomials is given by:

$$\mathcal{M}(\mathcal{G}) = \{s_1 s_2^3 s_3, s_1 s_2^2 s_3, s_1 s_2^2, s_1 s_2 s_3, s_1 s_2, s_1 s_3, s_1, s_2^2 s_3, s_2 s_3, s_2, s_3\}$$

The monomial $s_2^2 s_3$ and $s_1 s_2^2$ identify the confounded directed edge $X_3 \rightarrow X_2$ and $X_1 \rightarrow X_2$, respectively, and thus our algorithm correctly identifies the equivalence class of Type 20.

**Type 21** For all DAGs falling into this equivalence class, the set of distinct monomials is given by:

$$\mathcal{M}(\mathcal{G}) = \{s_1 s_2^2, s_1 s_2, s_1, s_2^2 s_3, s_2 s_3, s_2, s_3\}$$

The monomials $s_1 s_2^2$ and $s_2^2 s_3$ identify the confounded directed edges $X_1 \rightarrow X_2$ and $X_3 \rightarrow X_2$, respectively, via Theorem 3.8. Consequently, our algorithm correctly identifies the equivalence class of Type 21.

**Type 22** For all DAGs falling into this equivalence class, the set of distinct monomials is given by:

$$\mathcal{M}(\mathcal{G}) = \{s_1 s_2^2 s_3, s_1 s_2^2, s_1 s_2 s_3, s_1 s_2, s_1 s_3, s_1, s_2^2 s_3, s_2 s_3, s_2, s_3\}$$

The monomials $s_1 s_2^2$ and $s_2^2 s_3$ identify the confounded directed edges $X_1 \rightarrow X_2$ and $X_3 \rightarrow X_2$ via Theorem 3.8. The remaining edge $X_1 \leftrightarrow X_3$ is identified as bi-directed due to the absence of $s_1 s_2 s_3^3$ by Type (f3) in Theorem 3.12. Consequently, our algorithm correctly identifies the equivalence class of Type 22.

**Type 23**  For all DAGs falling into this equivalence class, the set of distinct monomials is given by:

$$\mathcal{M}(\mathcal{G}) = \{s_1^3 s_2^2 s_3, s_1^3 s_2 s_3, s_1^2 s_2^2 s_3, s_1^2 s_2 s_3, s_1^2 s_2, s_1^2 s_3, s_1 s_2^2 s_3, s_1 s_2 s_3, s_1 s_2, s_1 s_3, s_1, s_2^2 s_3, s_2 s_3, s_2, s_3\}$$

This set provides comprehensive identification: $s_1^2 s_2$ identifies $X_2 \rightarrow X_1$, $s_1^2 s_3$ identifies $X_3 \rightarrow X_1$, and $s_2^2 s_3$ identifies $X_3 \rightarrow X_2$. All three are confounded directed edges identified via Theorem 3.8. Consequently, our algorithm correctly identifies the equivalence class of Type 23.

**Type 24**  For all DAGs falling into this equivalence class, the set of distinct monomials is given by:

$$\mathcal{M}(\mathcal{G}) = \{s_1^4 s_2^2 s_3, s_1^3 s_2^2 s_3, s_1^3 s_2 s_3, s_1^2 s_2^2 s_3, s_1^2 s_2 s_3, s_1^2 s_2, s_1^2 s_3, s_1 s_2^2 s_3, s_1 s_2 s_3, s_1 s_2, s_1 s_3, s_1, s_2^2 s_3, s_2 s_3, s_2, s_3\}$$

Similar to Type 23, the monomials $s_1^2 s_2$, $s_1^2 s_3$, and $s_2^2 s_3$ identify the three confounded directed edges $X_2 \rightarrow X_1$, $X_3 \rightarrow X_1$, and $X_3 \rightarrow X_2$ via Theorem 3.8. The specific higher-order terms like $s_1^4 s_2^2 s_3$ distinguish this class's underlying constraints. Consequently, our algorithm correctly identifies the equivalence class of Type 24.

**Type 25**  For all DAGs falling into this equivalence class, the set of distinct monomials is given by:

$$\mathcal{M}(\mathcal{G}) = \{s_1^3 s_2 s_3^2, s_1^2 s_2 s_3^2, s_1^2 s_2 s_3, s_1 s_2 s_3^2, s_1 s_2 s_3, s_1 s_2, s_1 s_3, s_1, s_2 s_3^2, s_2 s_3, s_2, s_3\}$$

The monomial $s_2 s_3^2$ identifies the confounded directed edge $X_2 \rightarrow X_3$ via Theorem 3.8. The higher-order term $s_1^3 s_2 s_3^2$ identifies the collider pattern $X_2 \rightarrow X_1$ and $X_3 \rightarrow X_1$ via Type (d2) in Theorem 3.9. Consequently, our algorithm correctly identifies the equivalence class of Type 25.

**Type 26**  For all DAGs falling into this equivalence class, the set of distinct monomials is given by:

$$\mathcal{M}(\mathcal{G}) = \{s_1^3 s_2 s_3, s_1^2 s_2 s_3, s_1^2 s_2, s_1 s_2 s_3, s_1 s_2, s_1 s_3, s_1, s_2 s_3, s_2, s_3\}$$

The monomial $s_1^2 s_2$ identifies the confounded directed edge $X_2 \rightarrow X_1$ via Theorem 3.8. The remaining edges $X_3 \rightarrow X_1$ and $X_3 \rightarrow X_2$ are identified as directed through the high-order structural implications of $s_1^3 s_2 s_3$ via Type (d3) in Theorem 3.10, matching the full DAG structure. Consequently, our algorithm correctly identifies the equivalence class of Type 26.

**Type 27**  For all DAGs falling into this equivalence class, the set of distinct monomials is given by:

$$\mathcal{M}(\mathcal{G}) = \{s_1^3 s_2 s_3^2, s_1^3 s_2 s_3, s_1^2 s_2 s_3^2, s_1^2 s_2 s_3, s_1^2 s_2, s_1 s_2 s_3^2, s_1 s_2 s_3, s_1 s_2, s_1 s_3, s_1, s_2 s_3^2, s_2 s_3, s_2, s_3\}$$

This set identifies two confounded directed edges: $X_2 \rightarrow X_1$ (via $s_1^2 s_2$) and $X_2 \rightarrow X_3$ (via $s_2 s_3^2$). The edge $X_3 \rightarrow X_1$ is identified via Type (d3) in Theorem 3.10. Consequently, our algorithm correctly identifies the equivalence class of Type 27.

**Type 28**  For all DAGs falling into this equivalence class, the set of distinct monomials is given by:

$$\mathcal{M}(\mathcal{G}) = \{s_1, s_2, s_3\}$$

The absence of any joint monomials (e.g., $s_i s_j$) indicates that there are no edges of any kind between the variables. Consequently, our algorithm correctly identifies the equivalence class of Type 28.

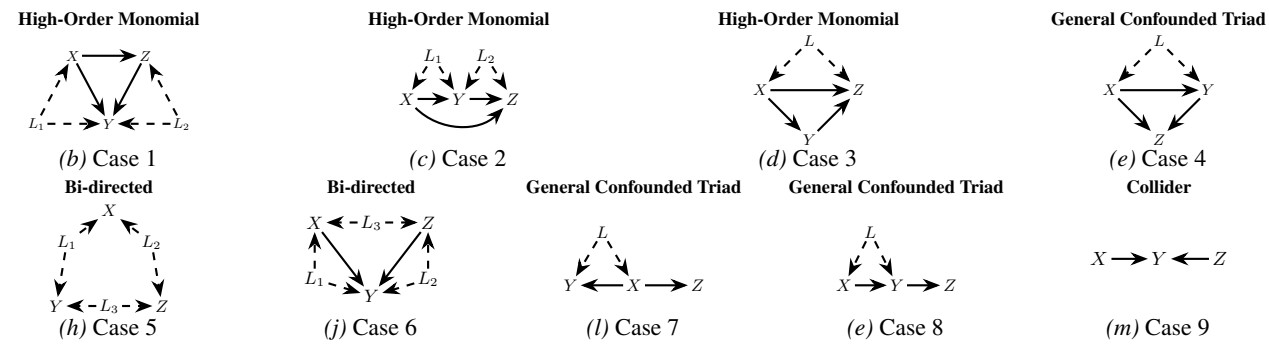

*Figure 9.* Overview of 9 causal structures in additional experiments. Nodes are observed variables ($X, Y, Z$) and latent variables ($L$).

**Type 29** For all DAGs falling into this equivalence class, the set of distinct monomials is given by:

$$\mathcal{M}(\mathcal{G}) = \{s_1 s_2, s_1 s_3, s_1, s_2 s_3, s_2, s_3\}$$

The set identifies adjacencies between all pairs ($s_1 s_2, s_1 s_3, s_2 s_3$). The complete absence of higher-order directionality monomials implies that all edges are bi-directed ($X_1 \leftrightarrow X_2, X_1 \leftrightarrow X_3, X_2 \leftrightarrow X_3$) by Theorem 3.12. Consequently, our algorithm correctly identifies the equivalence class of Type 29.

In summary, by analyzing the overall 29 equivalence classes, we show that every edge with arrow head are indeed identifiable by one of the proposed theorems, and thus we can conclude that Theorems 3.7 to 3.12 are complete for the local 3-variables identification in LC-PB-SCM.

$\square$

# C. Additional Experiments

## C.1. Additional Synthetic Experiments

Extra cases are present in Figure 9. Performances are given in Table 2. The LC-PB-SCM demonstrates consistent performance gains as the sample size increases. Specifically, F1 scores improve from an initial range of 0.78–0.95 with 5,000 samples to 0.95–1.00 with 100,000 samples. Concurrently, the standard deviation decreases substantially, underscoring the method's enhanced stability and reliability. Notably, even in Case 9, where FCI excels, our method still achieves comparable results, thereby highlighting its robustness across diverse settings. FCI exhibits weak performance (with scores approaching zero) in Cases 1–8. This limitation stems from the fact that FCI is restricted to identifying the skeleton and v-structures within the Markov Equivalence Class, failing to fully recover the underlying ground truth causal structure.

Notably, in the case study, confirming the ground truth PPADMG required us to exhaustively explore all possible DAGs sharing the same PGF as the underlying case DAG. However, this brute-force approach is computationally intensive and cannot be scaled to scenarios with a larger number of variables. Nevertheless, this limitation applies only to the validation phase and does not restrict the applicability of our method to more complex structures.

To illustrate this, we evaluated our method on synthetic data generated from random graphs consisting of 10 variables(with 6 observed variables and 4 latent confounders) and 20 variables(with 12 observed variables and 8 latent confounders) across varying sample sizes. The default settings are as follows: range of causal coefficient $\alpha_{i,j} \in [0.1, 0.5]$, range of the mean Poisson noise $\mu_i \in [0.001, 0.005]$. Each simulation is repeated 50 times. Note that we utilize a modified $F_1$ score metric where the uncertainty mark "∘" is considered compatible with specific edge marks. Besides, comparisons are made directly against the ground truth PADMG rather than PPADMG. Due to the limited performance of OCD and its higher computational cost, the results of OCD are not included in the comparison of 20 variables. Performances are given in Table 3 and 4.

LC-PB-SCM consistently demonstrates superior performance, maintaining a clear lead over all baseline algorithms across the entire range of sample sizes in 10 variables experiment, while it still preserves its performance with high sample size in

*Table 2.* Performance comparison across extra Cases.

| Case | Size | LC-PB-SCM | FCI | PBSCM | GES | OCD |
|---|---|---|---|---|---|---|
| **Case 1** | 5k | **0.87**±**0.15** | 0.05±0.08 | 0.42±0.03 | 0.40±0.01 | 0.46±0.01 |
| | 10k | **0.89**±**0.15** | 0.05±0.08 | 0.42±0.00 | 0.40±0.01 | 0.46±0.03 |
| | 50k | **0.98**±**0.09** | 0.12±0.08 | 0.42±0.00 | 0.42±0.02 | 0.38±0.07 |
| | 100k | **1.00**±**0.00** | 0.15±0.05 | 0.42±0.00 | 0.42±0.02 | 0.33±0.02 |
| **Case 2** | 5k | **0.80**±**0.11** | 0.00±0.00 | 0.46±0.04 | 0.43±0.02 | 0.34±0.01 |
| | 10k | **0.78**±**0.10** | 0.00±0.00 | 0.46±0.06 | 0.44±0.01 | 0.33±0.01 |
| | 50k | **0.92**±**0.10** | 0.00±0.00 | 0.47±0.00 | 0.44±0.00 | 0.33±0.04 |
| | 100k | **0.95**±**0.08** | 0.00±0.00 | 0.47±0.00 | 0.44±0.00 | 0.33±0.00 |
| **Case 3** | 5k | **0.95**±**0.09** | 0.00±0.00 | 0.44±0.08 | 0.53±0.04 | 0.47±0.08 |
| | 10k | **0.99**±**0.04** | 0.00±0.00 | 0.47±0.00 | 0.56±0.00 | 0.48±0.10 |
| | 50k | **1.00**±**0.00** | 0.00±0.00 | 0.47±0.00 | 0.56±0.00 | 0.52±0.07 |
| | 100k | **1.00**±**0.00** | 0.00±0.00 | 0.47±0.00 | 0.56±0.00 | 0.55±0.03 |
| **Case 4** | 5k | **0.92**±**0.12** | 0.00±0.00 | 0.41±0.07 | 0.56±0.00 | 0.39±0.10 |
| | 10k | **0.96**±**0.10** | 0.00±0.00 | 0.42±0.06 | 0.56±0.00 | 0.39±0.11 |
| | 50k | **0.99**±**0.04** | 0.00±0.00 | 0.45±0.05 | 0.56±0.00 | 0.35±0.06 |
| | 100k | **1.00**±**0.02** | 0.00±0.00 | 0.46±0.03 | 0.56±0.00 | 0.34±0.04 |
| **Case 5** | 5k | **0.87**±**0.08** | 0.00±0.00 | 0.48±0.13 | 0.28±0.04 | 0.76±0.05 |
| | 10k | **0.88**±**0.18** | 0.00±0.00 | 0.46±0.15 | 0.30±0.04 | 0.77±0.00 |
| | 50k | **1.00**±**0.01** | 0.00±0.00 | 0.49±0.12 | 0.33±0.01 | 0.77±0.00 |
| | 100k | **1.00**±**0.00** | 0.00±0.00 | 0.45±0.15 | 0.33±0.00 | 0.77±0.00 |
| **Case 6** | 5k | **0.90**±**0.04** | 0.00±0.00 | 0.50±0.00 | 0.44±0.00 | 0.55±0.14 |
| | 10k | **0.92**±**0.06** | 0.00±0.00 | 0.50±0.00 | 0.44±0.00 | 0.58±0.13 |
| | 50k | **0.98**±**0.05** | 0.00±0.00 | 0.50±0.00 | 0.44±0.00 | 0.67±0.00 |
| | 100k | **1.00**±**0.00** | 0.00±0.00 | 0.50±0.00 | 0.44±0.00 | 0.67±0.00 |
| **Case 7** | 5k | **0.78**±**0.13** | 0.00±0.00 | 0.40±0.02 | 0.70±0.00 | 0.68±0.11 |
| | 10k | **0.85**±**0.16** | 0.00±0.00 | 0.40±0.02 | 0.70±0.02 | 0.70±0.10 |
| | 50k | **0.99**±**0.06** | 0.00±0.00 | 0.44±0.04 | 0.70±0.00 | 0.72±0.09 |
| | 100k | **0.99**±**0.05** | 0.00±0.00 | 0.46±0.04 | 0.70±0.02 | 0.72±0.09 |
| **Case 8** | 5k | **0.74**±**0.15** | 0.00±0.00 | 0.42±0.06 | 0.70±0.03 | 0.68±0.11 |
| | 10k | **0.82**±**0.18** | 0.00±0.00 | 0.41±0.05 | 0.70±0.02 | 0.70±0.10 |
| | 50k | **0.99**±**0.06** | 0.00±0.00 | 0.53±0.08 | 0.70±0.00 | 0.72±0.09 |
| | 100k | **1.00**±**0.00** | 0.00±0.00 | 0.57±0.03 | 0.70±0.00 | 0.72±0.09 |
| **Case 9** | 5k | 0.87±0.20 | **0.99**±**0.09** | 0.64±0.11 | 0.74±0.08 | 0.63±0.09 |
| | 10k | 0.98±0.09 | **1.00**±**0.00** | 0.65±0.09 | 0.77±0.04 | 0.61±0.09 |
| | 50k | **1.00**±**0.00** | **1.00**±**0.00** | 0.67±0.07 | 0.78±0.00 | 0.57±0.04 |
| | 100k | **1.00**±**0.00** | **1.00**±**0.00** | 0.68±0.04 | 0.78±0.00 | 0.56±0.01 |

20 variables experiment. These experimental results validate the method's applicability to multivariate settings.

Table 3. Performance comparison of random graphs consisting of 10 variables. Metrics are reported as F1..

| Sample Size | LC-PB-SCM | FCI | PBSCM | GES | OCD |
|---|---|---|---|---|---|
| 5000 | **0.60±0.06** | 0.58±0.08 | 0.06±0.09 | 0.38±0.07 | 0.17±0.07 |
| 10000 | **0.68±0.06** | 0.59±0.09 | 0.24±0.10 | 0.44±0.10 | 0.20±0.09 |
| 30000 | **0.73±0.05** | 0.61±0.09 | 0.45±0.08 | 0.48±0.09 | 0.22±0.08 |
| 50000 | **0.75±0.06** | 0.63±0.09 | 0.50±0.06 | 0.50±0.11 | 0.35±0.06 |
| 100000 | **0.77±0.06** | 0.64±0.08 | 0.57±0.04 | 0.52±0.08 | 0.28±0.14 |

Table 4. Performance comparison of random graphs consisting of 20 variables. Metrics are reported as F1.

| Sample Size | LC-PB-SCM | FCI | PB-SCM | GES |
|---|---|---|---|---|
| 5000 | 0.50±0.07 | **0.58±0.06** | 0.11±0.05 | 0.34±0.09 |
| 10000 | 0.59±0.06 | **0.62±0.06** | 0.23±0.06 | 0.40±0.09 |
| 30000 | **0.66±0.06** | 0.65±0.06 | 0.35±0.06 | 0.48±0.08 |
| 50000 | **0.70±0.05** | 0.67±0.06 | 0.40±0.04 | 0.51±0.09 |
| 100000 | **0.71±0.07** | 0.68±0.06 | 0.46±0.04 | 0.53±0.10 |

## C.2. Additional Real World Experiment on Football Events Dataset

We utilize a real-world football events dataset[3], comprising 941,009 events from 9,074 matches across major European leagues to validate our method. This experiment investigates the causal mechanisms among specific game events, namely *Foul*, *Yellow Card*, *Second Yellow Card*, *Red Card*, and *Substitution*, as illustrated in Figure 10. In our modeling setup, we designate the *Foul* event as a latent variable, while the remaining events are treated as observed variables. Consequently, the objective is to recover the underlying causal structure solely from the observed count data. Actually, we perform structure

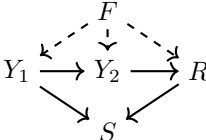

Figure 10. F: Foul, Y1: Yellow card, Y2: Second yellow card, R: Red card, S: Substitution, H: Hand ball

learning on the skeleton of the true causal graph, and our method successfully recovers the correct edge orientations of $Y_1 \rightarrow Y_2, Y_2 \rightarrow R$ by the theorem 3.8, $Y_1 \rightarrow S$, $R \rightarrow S$ by theorem 3.11.

## D. Additional Discussion of Complexity

Let $d$ denote the number of observed variables. Our algorithm has three phases: pairwise skeleton tests $O(d^2)$, confounded-edge detection $O(d^2)$, and triple orientation $O(d^3)$, yielding an overall complexity of $O(d^3)$. The overall complexity compared with the existing methods is listed in Table 5:

While our method achieves polynomial complexity, FCI and GES rely on conditional independence testing or equivalence class search, respectively, whose cost can grow exponentially with $d$ in dense graphs. OCD uses iterative ordinal regression, where $q$ is the number of nodes, $m$ is the maximum number of parents, and $L$ is the number of ordinal levels.

## E. Additional Discussion of Robustness

Specifically, we discuss the numerical stability of empirical log-PGF estimation in this section. We use the empirical log-PGF as a consistent estimator for evaluating the true log-PGF at given points. To illustrate the gap between the empirical PGF and the true PGF, we consider a single random variable $X$ as an example, and the estimation error can be formalized as

---

[3]https://www.kaggle.com/datasets/secareanualin/football-events

*Table 5.* Computational complexity comparison of different methods.

| Method | Complexity |
|---|---|
| LC-PB-SCM (Ours) | $O(d^3)$ |
| PB-SCM | $O(d^3)$ |
| FCI | Exponential (worst case) |
| GES | Exponential (worst case) |
| OCD | $O(q \cdot f(n, m, L))$ per iteration |

$\hat{G}_n(s_0) - G_X(s_0) = \frac{1}{n} \sum_{i=1}^{n} s_0^{x_i} - \mathbb{E}\left[s_0^X\right]$ where $G_X(s) = \sum_{x=0}^{\infty} p(x)s^x = \mathbb{E}[s^X]$ is the true PGF of the random variable $X$, and $\hat{G}_n(s) = \frac{1}{n} \sum_{i=1}^{n} s^{x_i}$ is the sample mean of $s^X$, i.e., the empirical PGF. According to the *Central Limit Theorem*, we have $\sqrt{n}\left(\hat{G}_n(s_0) - G_X(s_0)\right) \xrightarrow{d} \mathcal{N}(0, \sigma^2)$, where $n$ is the sample size and $\sigma^2 = \mathrm{Var}(s_0^X) = \mathbb{E}[s_0^{2X}] - \mathbb{E}[s_0^X]^2 = G_X(s_0^2) - [G_X(s_0)]^2$. A naive idea is to increase the sample size $n$ to reduce the variance.

To investigate how sample size $n$ affects the results with empirical log-PGF estimation, we design two complementary experiments: one evaluates the accuracy of empirical PGF estimation itself, while the other assesses the precision of coefficient estimation for individual monomials in the log-PGF expansion. We consider a simple causal structure $X_1 \to X_2$ with latent confounding $X_1 \leftarrow L \to X_2$ such that $\alpha_{1,2} = 0.3, \alpha_{L,1} = 0.4, \alpha_{L,2} = 0.5$, and $\mu = (0.06, 0.07, 0.08)$. The corresponding log-PGF is given as follows:

$$\log G_{\mathbf{X}}(\mathbf{s}) = 0.0048 s_1 s_2^2 + 0.034 s_1 s_2 + 0.0532 s_1 + 0.094 s_2 - 0.186.$$

We first examine the convergence of the empirical PGF $\hat{G}_{\mathbf{X}}(\mathbf{z})$ to its true value. At the evaluation point $\mathbf{z} = (0.01, 0.01)$, the true value is $G_{\mathbf{X}}(\mathbf{z}) = 0.8315$. Table 6 reports the mean and standard deviation over 50 independent trials across varying sample sizes.

*Table 6.* Empirical PGF $\hat{G}_{\mathbf{X}}(\mathbf{z})$ with true value 0.8315.

| Sample Size | Mean | Std |
|---|---|---|
| 1000 | $8.3103 \times 10^{-1}$ | $1.2303 \times 10^{-2}$ |
| 5000 | $8.3163 \times 10^{-1}$ | $5.1854 \times 10^{-3}$ |
| 10000 | $8.3152 \times 10^{-1}$ | $3.4680 \times 10^{-3}$ |
| 50000 | $8.3161 \times 10^{-1}$ | $1.8911 \times 10^{-3}$ |
| 100000 | $8.3165 \times 10^{-1}$ | $8.2312 \times 10^{-4}$ |

Next, we focus on the monomial $s_1 s_2^2$, whose true coefficient in $\log G_{\mathbf{X}}$ is 0.0048. This coefficient corresponds to a specific partial derivative of the log-PGF, which we estimate empirically. Table 7 summarizes the estimation performance across sample sizes.

*Table 7.* Empirical log-PGF partial derivative with true value $4.8 \times 10^{-3}$.

| Sample Size | Mean | Std |
|---|---|---|
| 1000 | $4.5810 \times 10^{-3}$ | $2.9938 \times 10^{-3}$ |
| 5000 | $4.9668 \times 10^{-3}$ | $1.2516 \times 10^{-3}$ |
| 10000 | $4.7914 \times 10^{-3}$ | $9.6206 \times 10^{-4}$ |
| 50000 | $4.6099 \times 10^{-3}$ | $3.4061 \times 10^{-4}$ |
| 100000 | $4.7836 \times 10^{-3}$ | $2.8608 \times 10^{-4}$ |

These results demonstrate that while small sample sizes induce relatively large variance, which may potentially affect structural discovery decisions, the variance diminishes rapidly as $n$ increases. Notably, when $n \geq 10,000$, the standard deviation falls below the magnitude of the estimated mean, indicating stable and reliable inference.

## F. Additional Discussion of Binomial Thinning in the Presence of Latent Variables

To show the property, we consider the simple scenario where an exogenous latent variable $L$ affects only a single observed variable, i.e., $L \xrightarrow{\beta} X \xrightarrow{\alpha} Y$. It is straightforward to show that the influence of $L$ is absorbed into the noise term of $X$. Specifically, this structure is statistically equivalent to a model $\hat{X} \xrightarrow{\alpha} Y$ with an adjusted noise parameter $\hat{\mu}_X = \mu_L \beta + \mu_X$. This implies that the existence of such a single-child latent variable cannot be verified from observational data.

Next, consider a latent variable $L$ acting as a mediator between $X$ and $Y$: $X \xrightarrow{\alpha} L \xrightarrow{\beta} Y$. By analyzing the Probability Generating Function (PGF), one could find that this structure admits an equivalent representation $X \xrightarrow{\alpha \cdot \beta} \hat{Y}$, where the noise of $\hat{Y}$ is adjusted to $\hat{\mu}_Y = \mu_L \beta + \mu_Y$. Here, the latent node $L$ is effectively marginalized out, and its causal effect is consolidated into the direct edge from $X$ to $Y$. In essence, a latent variable with an out-degree of one is statistically redundant for the marginal distribution, as its effect is fully absorbed by the downstream variable. This allows us to simplify the causal graph by removing such latent nodes while maintaining identical statistical properties. The following theorem establishes the validity of this simplification.

**Theorem F.1** (Latent Variable Reduction). *Let $\mathbf{X} = \{X_1, \ldots, X_d\}$ is the random variables in a PB-SCM whose causal structure is given by a DAG $\mathcal{G} = (\mathbf{V}, \mathbf{E})$. Let $L \in \mathbf{V}$ denote a variables and consider $L$ having a single child $Y$ (i.e., $Ch(L) = \{Y\}$). Let $W := Pa(Y) \backslash L$. Suppose $Pa(L) \cap W = \varnothing$. That is, no parent of $L$ is a direct parent of $Y$ (i.e., no multi-edge). Then the marginal distribution on the remaining variables $\mathbf{X} \backslash \mathbf{X}_L$ is statistically equivalent to another PB-SCM model whose causal structure $\mathcal{G}' = (\mathbf{V}', \mathbf{E}')$ is given by:*

- *Removing the node $L$ and the edge $X_L \to Y$,*

- *For each $i \in Pa(L)$, replace edge $X_i \to X_L$ by a direct edge $X_i \to Y$ with coefficient $\alpha'_{i,L} := \alpha_{i,L} \alpha_{L,Y}$,*

- *Update only the Poisson mean at $Y$ by $\mu'_Y := \mu_Y + \alpha_{L,Y} \mu_L$,*

- *Leaving all other edges (and their coefficients) unchanged.*

*Proof.* We first introduce an important property of the thinning operator on PGF for our proof.

**Property F.2** (Binomial Thinning on PGF). Let $Y = \rho \circ X$, where $\circ$ denotes the binomial thinning operator and parameter $\rho \in (0, 1)$. If $G_X(s)$ is the PGF of $X$, then the PGF of $Y$ is given by $G_Y(s) = G_X(\rho s + 1 - \rho)$. More generally, if a variable $X_i$ contributes to $X_j$ via thinning $\rho$, this corresponds to a variable substitution in the PGF domain:

$$s_i \mapsto s_i(\rho s_j + (1 - \rho)).$$

Now we start to prove theorem F.1 by the theorem A.3 and the property F.2, 2.6.

Let $\phi_i(\mathbf{s})$ be the function as define at A.2. Consider the $\phi_L(\mathbf{s})$ after marginalizing out $L$:

$$\phi_L(\mathbf{s} \backslash \{s_L\}, s_L = 1) = s_L(\alpha_{L,Y} \phi_Y(\mathbf{s}) + 1 - \alpha_{L,Y})$$
$$= \alpha_{L,Y} \phi_Y(\mathbf{s}) + 1 - \alpha_{L,Y}.$$

Therefore, the exponent of the PGF becomes

$$\mu_L(\phi_L(\mathbf{s} \backslash \{s_L\}, s_L = 1) - 1) = \mu_L \alpha_{L,Y}(\phi_Y(\mathbf{s}) - 1),$$

which can be absorbed into the Poisson noise of $Y$ as the parameter update $\mu'_Y := \mu_Y + \alpha_{L,Y} \mu_L$.

Next, consider any $i \in Pa(L)$, the $\phi_i(\mathbf{s})$ becomes

$$s_i(\alpha_{i,L} \alpha_{L,Y} \phi_Y(\mathbf{s}) + 1 - \alpha_{i,L} \alpha_{L,Y})$$

by the property F.2 and 2.6. This expression is exactly the $\phi_i(\mathbf{s})$ that performs thinning to $Y$ with coefficient $\alpha'_{i,L} := \alpha_{i,L} \alpha_{L,Y}$.

All other factors in $\phi_i(\mathbf{s})_{i \in \mathbf{V} \backslash \{Pa(L), L, Y\}}$ are unchanged. Consequently, the collection $\{\phi_i(\mathbf{s} \backslash \{s_L\}, s_L = 1)\}_{i \in \mathbf{V} \backslash L}$ satisfies the same recursion as $\{\phi'_i(\mathbf{s})\}_{i \in \mathbf{V}'}$ on the reduced DAG $\mathcal{G}'$, and hence: $\phi_i(\mathbf{s}) \equiv \phi'_i(\mathbf{s})$ for all $i \in \mathbf{V}'$.

Then we can yield the marginal global PGF

$$G_{\mathbf{X} \setminus X_L}(\mathbf{s} \setminus \{s_L\}, \ s_L = 1) = \exp \sum_{i \in \mathbf{V}'} \mu_i'(\phi_i'(\mathbf{s}))$$

which proves the claim. □

### F.1. Proof of Property F.2

*Proof.* Let $G_{\mathbf{X}}(\mathbf{s}) = \mathbb{E}[\mathbf{s}^{\mathbf{X}}]$ denote the Probability Generating Function (PGF) of the random vector $\mathbf{X}$. Consider a new variable $X_j$ generated from $X_i$ via a binomial thinning process, such that $X_j \mid X_i \sim \text{Binomial}(X_i, \rho)$. The joint PGF of the augmented set $\mathbf{X} \cup \{X_j\}$ can be derived as follows:

$$
\begin{aligned}
G_{\mathbf{X}, X_j}(\mathbf{s}, s_j) &= \mathbb{E}\left[\mathbf{s}^{\mathbf{X}} s_j^{X_j}\right] \\
&= \mathbb{E}_{\mathbf{X}}\left[\mathbf{s}^{\mathbf{X}} \cdot \mathbb{E}_{X_j \mid X_i}\left[s_j^{X_j} \mid X_i\right]\right] \\
&= \mathbb{E}_{\mathbf{X}}\left[\mathbf{s}^{\mathbf{X}}(\rho s_j + 1 - \rho)^{X_i}\right] \\
&= \mathbb{E}_{\mathbf{X}}\left[\prod_{k \neq i} s_k^{X_k} \cdot s_i^{X_i}(\rho s_j + 1 - \rho)^{X_i}\right] \\
&= G_{\mathbf{X}}\left(s_1, \ldots, s_i(\rho s_j + 1 - \rho), \ldots, s_d\right).
\end{aligned}
\tag{F.1}
$$

This demonstrates that the thinning operation corresponds to a specific argument substitution in the PGF domain. □

