# OpenReview forum: "On the Identifiability of Poisson Branching Structural Causal Model Under Latent Confounding"
_ICML.cc/2026/Conference — ICML 2026 spotlight_

### Official Review · Reviewer_b1J4 · 2026-03-08

**Soundness:** 2
**Presentation:** 2
**Significance:** 3
**Originality:** 3
**Overall Recommendation:** 5
**Confidence:** 3

**Summary:**

The paper addresses identifiability of SCMs based on Poisson variables when assuming possible existence of latent confounders. This is done by exhaustively proving identifiability for multiple possible scenarios in a base model involving at most three observed variables and two latent confounders using a result connecting 'Trie' graphical structures to observed monomials in the Poisson PGF. An algorithm is provided to identify these structures and is then applied to synthetic and real data with confounders.

**Compliance With Llm Reviewing Policy:**

Affirmed.

**Final Justification:**

The work meaningfully extends existing work to the case of latent confounders and provides comprehensive evaluation of the scenarios covered by the framework. My main concern was the presence of multiple mathematical typos and unclear notations/definitions that can greatly hamper a work emphasizing theoretical contributions. However, the authors clarified the main definitions and acknowledged they will fix the typos.

**Key Questions For Authors:**

1. Please rewrite Definition 3.1 without using unintroduced variables/notation and either create a Trie example or point to specific examples of Tries in Figure 2. This will help clarify a critical definition in this work.
2. Please enumerate all possible equivalence classes given up to three observed variables and two latents and connectivity types from Definition 2.3. For example, none of the theorems seem to subsume the case where a latent confounder affects X1 and X2, but X1 and X2 are neither connected by an arrowhead nor a circle. For instances not subsumed by your theorems, please discuss why identifiability doesn’t hold in those cases (or why these cases weren’t included if they are identifiable). This will probe the edges of identifiability of LC-PB-SCM and show the necessity of proving all cases separately instead of one unified theorem.
3. Can you show the most typical (e.g., median over your 50 simulations) example of graphs learned by FCI, PB-SCM, GES, and OCD for the synthetic experiments? This will elucidate the types of edges other methods fail to pick up on and sharpen the strengths of LC-PB-SCM.

**Limitations:**

yes

**Strengths And Weaknesses:**

Strengths:
The paper builds naturally and meaningfully on an existing work (Xiang et al, 2024) by extending identifiability of Poisson branching SCMs to latent confounding by proposing LC-PB-SCM. After introducing a fundamental ‘Trie’ structure and relating it to the PGF, the authors comprehensively prove identifiability for various scenarios of connectivity. LC-PB-SCM is then shown to significantly improve on existing methods, including PB-SCM, across multiple cases and dataset sizes highlighting its robustness.

Weaknesses:
For a theoretical contribution, the work is peppered with an unusually large number of typos and confusing definitions, examples, and sentences that impede the reader’s ability to understand results and assess soundness. While these were only the points I caught, I highly encourage the authors to carefully read through all mathematical details in the paper to correct sloppy notation.
- Definition 3.1: “for every valid directed path pi starting at s”, what is s? Then, “obtained by appending a single vertex v to pi”, is v different from n_pi’?
- Authors never explicitly show an example in Figure 2 of what a Trie is (they mention “a Trie root at L”, but never a Trie.
- "Since all nodes in this tree are induced by directed” unfinished sentence.
- Definition 3.4: set definition for S(T_G(v_0)) is incomplete and unclear.
- Theorem 3.5: authors refer to a singular node “N*_T” but then define it as a set in m(T*). Furthermore, m(T*) is defined locally within the theorem, and reused again later in definition 3.6 where readers must infer m(T’) is the quantity defined in Theorem 3.5.
- In multiple places the authors refer to an index belonging to a node set, e.g. Definition 3.6 k\in V, Theorem D.1 “i\in Pa(L)”, Theorem D.1 “removing the node L and the edge X_L->Y” is unclear why they use X_L instead of L if they had defined L to be a node, Definition B.1 “Trie rooted at i” instead of v_i and j\in Ch_T(i). Is V a set of nodes or a set of indices?
- Theorem 3.7: use of M(G) instead of \mathcal{M}(G) as defined earlier.
- Figure for Theorem 3.12 (f3), are the two latents L meant to be the same variable? For example, Theorem 3.10 (d1) shows L1 and L2 are distinct.
- Theorem A.1: equation A.1 switches from z argument to s.
- Theorem A.2: defines Ch(i) = \phi (I assume this is supposed to be the null set, but quite confusing to read given there is a phi defined just before Ch(i)). Then “by combining Eq. A.5 and Eq. A.5”.
- PB-SCM is never defined in the paper even though the authors rely heavily on it, and it is left up to the reader to infer what it is from definition 2.1. For example, Theorem D.1 invokes PB-SCM assumptions but does not refer to any definition within the paper.
- Theorem D.1: “phi_L(s)” is an undefined quantity within the theorem, should be linked to equation A.3.
- Generally, it would be helpful to provide in the appendix a brief background on PGF for Poisson and explicit derivation of the motivating example claim in Sec. 3.1 “X_1->X_2 generates the term alpha_{1,2}s_1s_2”.

---

> ### Author Rebuttal · Authors · 2026-03-31
>
> > **W1-W3 and Q1**: Regarding the definition and the example for Trie structure.
>
> **A1:** We sincerely thank the reviewer for the careful reading. We appreciate the opportunity to clarify the definition and intuition behind our Trie construction, which is a critical component of this work.
> The core intuition behind defining a Trie node by its entire directed path $\pi$ (denoted $n_\pi$) instead of just the graph vertex is to ensure uniqueness, so that the same vertex may appear multiple times in a Trie when it is reachable via different directed paths from the root.
>
> For example, considering Trie $\mathcal T_{\mathcal G_2}(L)$ shown in Figure 2(b), vertex $X_2$ induces two distinct Trie nodes including $n_{(L,X_2)}$ (induced by path $L \to X_2$) and $n_{(L,X_1,X_2)}$ (induced by path $L \to X_1 \to X_2$). This property is a direct consequence of the PGF of PB-SCM, as it is a composition of multiple directed paths.
>
> To further clarify the relation between $v$ and $n_{\pi'}$ in the definition, we let $v=X_2$, $\pi = (L, X_1)$. There exists a directed path $\pi' = \pi \oplus v = (L, X_1, X_2)$ in graph $\mathcal G_2$ that can be obtained by appending $v$ to $\pi$. Then, by definition, this creates a directed edge $(n_{(L, X_1)}, n_{(L, X_1, X_2)})$ in the Trie (which is the edge $X_1\to X_2$ shown in Figure 2(b)). Thus $v$ is a graph vertex, while $n_{\pi'}$ is the path-based Trie node.
>
> We have corrected the typos ($s$ → $v_0$; missing word "paths") and will revise the Trie definition with these examples in the camera-ready version.
>
> > **W4:** Definition 3.4 $\mathbb S(\mathcal T_ \mathcal G(v_0))$ is unclear
>
> **A2:** Thanks for pointing out this issue. Essentially, $\mathbb S(\mathcal T_{\mathcal G}(v_0))$ enumerates every possible sub-tree that can be obtained by selecting any subset of edges $\mathcal E_{cut} \subseteq \mathbf E_T$ to prune such that $\mathbb {S}(\mathcal T_{\mathcal G}(v_0))=$ {$\mathcal T' \mid \text{for any } \mathcal E_{cut} \subseteq \mathbf E_{\mathcal T}$ }
>  where $\mathcal{T}'$ is the corresponding pruned Trie by each $\mathcal{E}_{cut}$.
> We will improve the clarity of Definition 3.4 in the revised version.
>
> > **W5-W7, W9-W13 :** The uncleanness and the inconsistent notation and typos
>
> **A3:** Thanks for the careful reading. All identified typos and notation inconsistencies will be corrected in the revised manuscript. In particular, we will refine the definition of the node set $\mathbf{N}_T^*$ in Theorem 3.5 and clarify the statement of $m(\mathcal{T}')$ in Definition 3.6. Furthermore, we will conduct a thorough consistency check of all mathematical notation and, as suggested, provide a more detailed introduction to PB-SCM.
>
> > **W8:** Figure for Theorem 3.12 (f3) ...
>
> **A4:** Thanks for identifying this potential source of confusion. Based on Theorem 3.12, the two latent variables $L$ are not required to be the same variable. To avoid ambiguity, we will update Figure (f3) in the revised manuscript to distinguish between these latents.
>
> > **Q2 :** Please enumerate ... if they are identifiable.
>
> **A5:** Thank you for this question.
> We have provided a complete enumeration in Appendix B.8 (Proof of Theorem 3.13), in which we systematically enumerate all possible structure and after accounting for isomorphism, we obtain exactly 29 unique monomial equivalence classes for local 3-variable LC-PB-SCMs (Figure 7).
>
> Regarding the case where a latent confounder affects $X_1$ and $X_2$, this specific case is indeed non-identifiable, and is addressed by Theorem 3.13. Specifically, this case falls into the same monomial equivalence class represented as $X_1 \circ-\circ X_2$ in the PPADMG (categorized under Type 1 in Figure 7). Further discussion is in the proof of Theorem 3.7 (Appendix B.2). We will include additional discussion of two-variable cases in the camera-ready version.
>
> > **Q3:** Can you show ... synthetic experiments?
>
> **A6:** We would like to take Figure 4(b) (Case 2 Collier) as an example, satisfying $X\leftarrow L\to Y$, and both $X \to Z \leftarrow Y$. The ground-truth of PPADMG of this case is: $X \circ-\circ Y$ and $X \circ\to Z \leftarrow \circ Y$.
>
> In general, our method successfully identify the ground-truth PPADMG, whereas the results of baseline models are listed as follows:
>
> - **FCI:** recovers the skeleton ($X \circ -\circ Y$, $X \circ - \circ Z$, $Y \circ - \circ Z$) but fails to orient any edges due to Markov equivalence class constraints.
> - **PB-SCM:** outputs $X \circ - \circ Y$, $X \to Z$, $Y \to Z$ but neglect the influence of latent confounding.
> - **GES:** yields a simple undirected triangle, failing to capture any causal directionality.
> - **OCD:** produces a reversed or incorrect orientation ($X \to Y, X \leftarrow Z, Y \leftarrow Z$) that does not align with the underlying data.
>
> This shows that LC-PB-SCM's higher-order PGF monomials enable accurate collider identification where standard methods fail. We will include this case study in the revised manuscript.

---

> > ### Author Rebuttal · Reviewer_b1J4 · 2026-04-02
> >
> > I thank the authors for their detailed response that resolves my concerns and am happy to raise my score.
> >
> > I strongly encourage the authors to carefully check the correctness of all mathematical statements in the next version of this paper.

---

### Official Review · Reviewer_V43M · 2026-03-11

**Soundness:** 3
**Presentation:** 3
**Significance:** 3
**Originality:** 3
**Overall Recommendation:** 5
**Confidence:** 2

**Summary:**

This paper investigates the causal structure learning on the Poisson Branching Structural Causal Model without the assumption of causal sufficiency. The identifiability property is built and the developed approach from the PGF can be utilized to identify the causal structure. Experimental results on both synthetic data and real-world datasets verify the correctness and effectiveness.

**Compliance With Llm Reviewing Policy:**

Affirmed.

**Final Justification:**

My concerns are addressed and I will keep the current ratings.

**Key Questions For Authors:**

1. What is the computational complexity of the proposed method, and can you show the cost comparison with existing methods?
2. Can you privide more discussions or comparisons with more existing work on causal discovery with latent confounding?

**Limitations:**

yes

**Strengths And Weaknesses:**

# Strengths

- Compared with existing work PB-SCM, this work extends it into a more general condition without the assumption of causal sufficiency. This generalization is meaningful and the motivation is clear.
- The writing is clear and well-organized.
- The experiments clearly support the proposals.

# Weakness
- This paper does not provide an analysis on its computational complexity and runtime cost comparison, which I believe is also valuable for an algorithm.
- The related work is not sufficiently discussed. I think there are many other methods that focus on causal discovery with latent confounding, while only several of them are mentioned in this paper. Despite the uniqueness of the count data, I am curious about and would be glad to see the discussions and comparisons with some other existing related methods, e.g.

[1] Differentiable Causal Discovery for Latent Hierarchical Causal Models. ICLR, 2025.
[2] Scalable Differentiable Causal Discovery in the Presence of Latent Confounders with Skeleton Posterior. KDD, 2024.

---

> ### Author Rebuttal · Authors · 2026-03-31
>
> > **Q1 :** What is the computational complexity of the proposed method, and can you show the cost comparison with existing methods?
>
> **A1 :** Thank you for raising this point. Let $d$ denote the number of observed variables. Our algorithm has three phases: pairwise skeleton tests $O(d^2)$, confounded-edge detection $O(d^2)$, and triple orientation $O(d^3)$, yielding an overall complexity of $O(d^3)$. The overall complexity compared with the existing methods is listed as follows:
>
> | Method | Complexity |
> |---|---|
> | **LC-PB-SCM (Ours)** | $O(d^3)$ |
> | **PB-SCM** | $O(d^3)$ |
> | **FCI** | Exponential (worst case) |
> | **GES** | Exponential (worst case) |
> | **OCD** | $O(q \cdot f(n,m,L))$ per iteration |
>
> While our method achieve polynomial complexity, FCI and GES rely on conditional independence testing or equivalence class search respectively, whose cost can grow exponentially with $d$ in dense graphs. OCD uses iterative ordinal regression, where $q$ is the number of nodes, $m$ the max parents, and $L$ the number of ordinal levels. Based on your suggestion, we would like to include the complexity analysis of our proposed method in the camera-ready version.
>
> > **Q2 :** Can you provide more discussions or comparisons with more existing work on causal discovery with latent confounding?
>
> **A2:** Thank you for your valuable suggestion. We would We discuss the related work on causal discovery with latent confounders below. We would love to provide a more comprehensive discussion and comparison with existing literature on causal discovery with latent confounders.
>
> In the presence of latent confounders, constraint-based methods such as FCI [1] leverage conditional independence tests to output partial ancestral graphs (PAGs). To go beyond Markov equivalence classes, a significant line of work introduces parametric assumptions. Early methods utilize rank conditions on covariance matrices to locate latent variables [2]. The Generalized Independent Noise (GIN) condition [3] extends this to linear non-Gaussian settings, which has inspired a series of remarkable subsequent works focusing on identifying linear [4, 5, 10] and nonlinear [6] latent hierarchical structures. Recently, these rank-based ideas have also been elegantly extended to discrete data via tensor rank conditions [7]. Alongside these developments, scalable differentiable methods have emerged for ADMG learning [8] and latent hierarchical discovery [9].
>
>
> While the aforementioned methods address general continuous or discrete data under linear or nonlinear functional relations, our work focus on a specific but commonly accounted count data setting that is generated by branching structures. We leverage the algebraic properties of PGFs to establish identifiability, which is complementary to the independence/rank-based tools used in existing work. We would also like to further explore the possible applicability of the current method especially the rank based method for discrete variable under the count data in the future direction. We would like to incorporate these expanded discussions into the camera-ready version.
>
> [1] Causation, Prediction, and Search. MIT Press, 2000.
>
> [2] Learning the Structure of Linear Latent Variable Models. JMLR, 2006.
>
> [3] Generalized Independent Noise Condition for Estimating Latent Variable Causal Graphs. NeurIPS, 2020.
>
> [4] Identification of Linear Non-Gaussian Latent Hierarchical Structure. ICML, 2022.
>
> [5] Latent Hierarchical Causal Structure Discovery with Rank Constraints. NeurIPS, 2022.
>
> [6] Identification of Nonlinear Latent Hierarchical Models. NeurIPS, 2023.
>
> [7] Learning Discrete Latent Variable Structures with Tensor Rank Conditions. NeurIPS, 2024.
>
> [8] Scalable Differentiable Causal Discovery in the Presence of Latent Confounders with Skeleton Posterior. KDD, 2024.
>
> [9] Differentiable Causal Discovery for Latent Hierarchical Causal Models. ICLR, 2025.
>
> [10] Some General Identification Results for Linear Latent Hierarchical Causal Structure. IJCAI, 2023.

---

> > ### Author Rebuttal · Reviewer_V43M · 2026-03-31
> >
> > Thank the authors for the response. My concerns are addressed.

---

### Official Review · Reviewer_7WtZ · 2026-03-12

**Soundness:** 2
**Presentation:** 3
**Significance:** 3
**Originality:** 4
**Overall Recommendation:** 4
**Confidence:** 3

**Summary:**

This paper investigates the challenging problem of causal discovery from observational count data in the presence of latent confounders. To model the underlying branching dynamics, the authors introduce a novel graphical representation called a Trie, which effectively maps causal mechanisms to the algebraic properties of Probability Generating Function (PGF) monomials. Based on this mapping, the paper establishes complete identifiability conditions for local $3$-variable substructures within the Latent Confounded Poisson Branching Structural Causal Model (LC-PB-SCM). Finally, the theoretical findings are translated into a practical algorithm for causal structure recovery, which is evaluated on both synthetic and real-world datasets.

**Compliance With Llm Reviewing Policy:**

Affirmed.

**Final Justification:**

Overall, I find the theory technically solid, and the methodology well motivated, so I maintain my weak accept recommendation.

**Key Questions For Authors:**

1. **Regarding the numerical stability of empirical log-PGF estimation (Addressing Soundness)**
   The theoretical mapping to PGF monomials is elegant, but estimating the empirical PGF $\mathbb{E}[s^X]$ from finite sample data can be notoriously unstable. Specifically, if the observational count values $X$ are large, the term $s^X$ can easily lead to severe variance, floating-point overflow, or underflow.
   - Question: How does your practical algorithm address this high variance and numerical instability during empirical log-PGF estimation? Are there specific constraints on the maximum count values in the datasets used?
   - How it changes my evaluation: If the authors can provide a clear explanation of their variance reduction techniques, numerical stabilization tricks, or theoretical bounds on the estimation error, it will significantly alleviate my concerns regarding the algorithm's real-world applicability, and I will gladly raise the "Soundness" score.
2. **Regarding scalability and error propagation from local to global graphs (Addressing Soundness & Significance)**
   The paper establishes a rigorous complete identifiability condition for local 3-variable structures. However, when building a global graph, local-to-global piece-together approaches are often highly vulnerable to error propagation (i.e., one misidentified local edge cascading through the entire network).
   - Question: How robust is the proposed global algorithm against this error accumulation, especially in denser or larger graphs? Could the authors provide additional empirical results or discussion on graphs with a larger number of nodes (e.g., $>20$ nodes)?
   - How it changes my evaluation: Demonstrating the algorithm's robustness on larger synthetic networks, or providing an explicit discussion on error propagation bounds, would convince me of the method's scalability, positively impacting my evaluation of both "Soundness" and "Significance".
3. **Regarding the assumption of mutually independent latent confounders (Addressing Soundness)**
   In Definition 2.1, the paper makes a strong assumption that all latent confounders are mutually independent root nodes. However, in many complex real-world systems, unobserved confounders are often causally related to one another (e.g., one latent confounder $L_1$ causes another latent confounder $L_2$, and both affect different observed variables).
   - Question: How sensitive is the proposed log-PGF monomial mapping to the violation of this independence assumption among latent variables? If latent confounders are dependent, would their correlation introduce unexpected cross-terms in the log-PGF expansion that could break the completeness of the identifiability condition for local 3-node structures?
   - How it changes my evaluation: I am not expecting a full theoretical proof for dependent confounders, as that might be beyond the scope of this paper. However, a clear theoretical discussion or a toy counter-example in the author's response clarifying whether the current Trie-to-monomial mapping breaks down under dependent latent variables would significantly improve my assessment of the model's theoretical boundaries and Soundness.

**Limitations:**

yes

**Strengths And Weaknesses:**

## Strengths
### Originality
The paper exhibits high originality by bridging causal graph topology with abstract algebra. Introducing the "Trie" representation to capture branching causal mechanisms and mapping them directly to the algebraic properties (monomials) of the log-Probability Generating Function (log-PGF) is a highly creative and novel approach. It provides a completely new "structural fingerprinting" perspective for causal discovery.
### Significance
Addressing latent confounding in observational count data is a notoriously difficult and highly relevant problem in many real-world domains (e.g., e-commerce, epidemiology). By relaxing the restrictive causal sufficiency assumption used in prior PB-SCM literature, this work unlocks new directions for causal inference on discrete branching dynamics, representing a meaningful advancement in the field.
### Soundness (Theoretical)
The theoretical foundation of the paper is exceptionally solid. The mathematical derivations linking causal paths to log-PGF expansion are rigorous. Furthermore, establishing a complete identifiability condition for local 3-variable substructures up to monomial equivalence is a strong theoretical contribution that robustly supports the paper's core claims.

### Presentation
The authors effectively use a motivating example (the Instrumental Variable scenario in Section 3.1) to introduce the core intuition before diving into heavy mathematical definitions. This significantly improves the readability of a highly theoretical paper.

## Weaknesses / Concerns
### Soundness (Empirical Estimation & Stability)
A critical concern lies in the empirical estimation of the log-PGF. In practice, estimating the PGF $E[s^X]$ from finite sample data can suffer from extreme variance and numerical instability, especially when the count values $X$ are large. The paper lacks a detailed discussion on how the proposed practical algorithm handles this numerical instability or bounds the estimation error, which may severely limit its real-world applicability on datasets with large counts.
### Soundness (Scalability & Error Propagation)
The theoretical identifiability is proven for local 3-variable structures. However, when the practical algorithm scales to larger graphs, such local-to-global approaches often suffer from severe error propagation (a single incorrect local edge identification cascades through the graph). The empirical evaluation does not sufficiently demonstrate the algorithm's robustness against this error accumulation on larger, denser networks (e.g., graphs with $>20$ nodes).
### Significance (Limitation of the Root-Node Assumption)
The authors assume that all latent confounders are mutually independent root nodes (Definition 2.1). While mathematically elegant (and distributionally equivalent to intermediate mediators as shown in the Appendix), collapsing latent mediators into direct edges removes the ability to identify intermediate variables.
### Presentation (Algorithmic Complexity)
While the theoretical mapping is clear, the computational complexity of enumerating and matching monomials in the empirical log-PGF expansion is not transparently discussed. A formal complexity analysis of the proposed algorithm with respect to the number of nodes and the maximum count values would improve the paper's completeness.

---

> ### Author Rebuttal · Authors · 2026-03-31
>
> > **Q1:**  How does your practical ... maximum count values in the datasets used?
>
> **A1:** Thanks for your enlightening comment.
>
> We would like to address this question by investigating how sample size $n$ affects the error of empirical log-PGF estimation. Specifically, we design two complementary experiments: one evaluates the accuracy of empirical PGF estimation itself, while the other assesses the precision of coefficient estimation for individual monomials in the log-PGF expansion.
> We consider a simple causal structure $X_1 \to X_2$ with latent confounding $X_1 \leftarrow L\to X_2$ such that $\alpha_{1,2} = 0.3,\alpha_{L,1} = 0.4,\alpha_{L,2} = 0.5$, and $\mathbf{\mu} = (0.06, 0.07, 0.08)$. The corresponding log-PGF is given as follows:
> $$\log G_{\mathbf{X}}(\mathbf{s}) = 0.0048s_1 s_2^2 + 0.034 s_1 s_2 + 0.0532 s_1 + 0.094 s_2 - 0.186$$
>
> We first examine the convergence of the empirical PGF
> $\hat G_\mathbf{X}(\mathbf{z})$
> to its true value. At the evaluation point $\mathbf{z} = (0.01, 0.01)$, the true value is $G_{\mathbf{X}}(\mathbf{z}) = 0.8315$.
> Table 1 reports the mean and standard deviation over 50 independent trials across varying sample sizes.
>
> **Table 1: Empirical PGF $\hat{G}_{\mathbf{X}}(\mathbf{z})$** (true value: $0.8315$)
> | Sample Size | Mean | Std |
> |---|---|---|
> | 5000 | $8.3163 \times 10^{-1}$ | $5.1854 \times 10^{-3}$ |
> | 10000 | $8.3152 \times 10^{-1}$ | $3.4680 \times 10^{-3}$ |
> | 50000 | $8.3161 \times 10^{-1}$ | $1.8911 \times 10^{-3}$ |
> | 100000 | $8.3165 \times 10^{-1}$ | $8.2312 \times 10^{-4}$ |
>
> Next, we estimate the coefficient of $s_1 s_2^2$ (true value: $0.0048$) via partial derivatives. Results are in Table 2.
>
> **Table 2: Empirical log-PGF partial derivative** (true value: $4.8 \times 10 ^{-3}$)
> | Sample Size | Mean | Std |
> |---|---|---|
> | 5000 | $4.9668 \times 10^{-3}$ | $1.2516 \times 10^{-3}$ |
> | 10000 | $4.7914 \times 10^{-3}$ | $9.6206 \times 10^{-4}$ |
> | 50000 | $4.6099 \times 10^{-3}$ | $3.4061 \times 10^{-4}$ |
> | 100000 | $4.7836 \times 10^{-3}$ | $2.8608 \times 10^{-4}$ |
>
> These results demonstrate that while small sample sizes induce relatively large variance which may potentially affecting structural discovery decisions, the variance diminishes rapidly as $n$ increases. Notably, when $n \geq 10,000$, the standard deviation falls below the magnitude of the estimated mean, indicating stable and reliable inference.
>
> Regarding the maximum count values, our method does not impose explicit hard constraints on the maximum count values in the datasets. However, we observe that data with smaller count ranges tend to exhibit lower variance, which facilitates more stable empirical log-PGF estimation. In practice, one may employ a bootstrap-based one-sided hypothesis test to ensure the robustness. We would like to include these discussions and experiments to improve the robustness of the proposed method.
>
> > **Q2**: How robust ...  (e.g., > 20 nodes)?
>
> **A2**: Thanks for your question. We agree that error propagation is a potential issue in our algorithm. To reduce its impact, we do not infer triple structures directly at the first stage in algorithm. Instead, we first identify the skeleton and subsequently the confounded-edge structure, since these rely on lower-order monomials and therefore on empirical PGF estimators with lower variance.
>
> To further demonstrate the algorithm's robustness on larger graphs, we conducted additional experiments on random graphs consisting of 12 observed variables and 8 latent confounders. The results are summarized below:
>
> | Sample Size | LC-PB-SCM | FCI | PB-SCM | GES |
> |---|---|---|---|---|
> | 5000 | $0.50 \pm 0.07$ | $\mathbf{0.58 \pm 0.06}$ | $0.11 \pm 0.05$ | $0.34 \pm 0.09$ |
> | 10000 | $0.59 \pm 0.06$ | $\mathbf{0.62 \pm 0.06}$ | $0.23 \pm 0.06$ | $0.40 \pm 0.09$ |
> | 30000 | $\mathbf{0.66 \pm 0.06}$ | $0.65 \pm 0.06$ | $0.35 \pm 0.06$ | $0.48 \pm 0.08$ |
> | 50000 | $\mathbf{0.70 \pm 0.05}$ | $0.67 \pm 0.06$ | $0.40 \pm 0.04$ | $0.51 \pm 0.09$ |
> | 100000 | $\mathbf{0.71 \pm 0.07}$ | $0.68 \pm 0.06$ | $0.46 \pm 0.04$ | $0.53 \pm 0.10$ |
>
> > **Q3:** How sensitive ... for local 3-node structures?
>
> **A3:** : Thank you for this insightful question. In our current framework, if this assumption is violated, for instance, if latent confounders are correlated or exhibit hierarchical dependencies, it would indeed introduce additional cross-terms in the log-PGF expansion, making the analysis substantially more complex. Addressing the full theoretical completeness of identifiability under dependent latents is a non-trivial extension. It would require characterizing the specific patterns of these unexpected cross-terms and how they aggregate across the network. We consider this an important direction for future work, where we aim to generalize the log-PGF expansion to accommodate more complex latent dependency structures. We would like to add some discussions under the violation of this independence assumption in the camera-ready version.

---

> > ### Author Rebuttal · Reviewer_7WtZ · 2026-04-02
> >
> > Thank you. I have no more concerns.

---

### Official Review · Reviewer_pbmQ · 2026-03-13

**Soundness:** 3
**Presentation:** 4
**Significance:** 3
**Originality:** 3
**Overall Recommendation:** 5
**Confidence:** 2

**Summary:**

This paper proves that the Poisson Branching SCM is identifiable under latent confounders. The paper connects the monomials of PGFs to branching causal mechanisms to establish an identifiability condition for local structure. The authors provide an algorithm for learning causal structures under latent confounding and demonstrate its effectiveness through a suite of synthetic and real-world experiments.

**Compliance With Llm Reviewing Policy:**

Affirmed.

**Final Justification:**

While I am not familiar with the causal inference literature, judging from the reviews of others and my initial reading of the manuscript, this paper provides a substantial theoretical and methodological contribution (extending the Poisson branching SBM to a setting with latent confounders). They provide many results on identifiability. While their experiments are initially somewhat limited, they provided a more extensive set of experiments in the rebuttal period.

**Key Questions For Authors:**

1. What is the limiting behavior of $G_X^L(s)$ in definition 2.5, as s goes to zero? Why is it not always zero? Could you provide intuition for this definition?
2. What are the most practical limitations of this method?
3. What do the results on the identifiability of local (3 node) structures imply for the identifiability of the global structure?

**Limitations:**

It is not clear to me how the method scales to settings with more than 5 variables, performs in small sample size settings, or how the local identifiability results extend to the global structure.

**Strengths And Weaknesses:**

Strengths:
* The theoretical results proving identifiability under latent confounding are strong.
* The algorithm yields compelling experimental results.
* The prose is clear and the paper well-written. The figures are nice and helpful for enhancing understanding.
* For a reader without a strong background in causal inference, section 3 is dense. However, it is not difficult to understand the rest of the paper even if omitting section 3, so this is not necessarily a problem.

Weaknesses:
* The experimental results have very few variables and may be considered toy examples. Scalability is a concern.

---

> ### Author Rebuttal · Authors · 2026-03-31
>
> > **Q1:** What is the limiting behavior of $G(s)$ in definition 2.5, as $s$ goes to zero? Why is it not always zero? Could you provide intuition for this definition?
>
> **A1:** Thank you for your comment. The reason for using the limitation is to handle the potential indeterminate form $0^0$, ensuring the definition is rigorous. Although it is possible to set $s$ to zero directly for some well-defined distributions, we still adopt the limit notation. This choice serves two purposes: on the one hand, it guarantees mathematical rigor; on the other hand, it maintains consistency in notation, since when we compute empirical probability generating functions, we cannot directly substitute $s = 0$ either. Intuitively, the local PGF is designed to simplify the analysis by focusing on local structures. The full joint PGF is a complex composite function (see Theorem A.2, Eq. A.3–A.4), and fully expanding it is intractable for estimation. Thus, by setting the non-target variables to zero, we retain only the monomials involving the target variables, effectively isolating the local causal structure of interest.
>
> > **Q2:** What are the most practical limitations of this method?
>
> **A2:** We appreciate the opportunity to clarify the practical boundaries of our method. First, the empirical PGF estimator is sensitive to sample size, and low-sample regimes may lead to higher variance, potentially affecting the robustness of monomial detection (see Reviewer 7WtZ Q1 for a detailed analysis). Second, our approach focuses on latent confounders within the PB-SCM framework. Expanding this to broader causal settings is an objective for future work. We would like to include the discussion of these practical considerations and future directions in the camera-ready version of the paper.
>
>
> > **Q3:** What do the results on the identifiability of local (3 node) structures imply for the identifiability of the global structure?
>
> **A3:** Thanks for your insightful question. Our algorithm reconstructs the global causal structure by systematically enumerating all local 3-variable substructures and applying the identifiability rules (Theorems 3.7–3.12). We also provide a theoretical guarantee (Theorem 3.13) that every 3-node subgraph is maximally identifiable by the proposed algorithm. For the global structure, it is possible to further identify the causal structure by exploiting additional information, e.g., structural priors or PGFs involving more than three variables.
> Extending identifiability to larger local substructures is a promising direction for future work. Nonetheless, even with only 3-node rules, our method achieves strong performance on larger graphs in practice (see Appendix C.1, Table 3 for 10-variable experiments). We would like to include a more detailed discussion of the implications of local identifiability for global structure in the camera-ready version of the paper.

---

> > ### Author Rebuttal · Reviewer_pbmQ · 2026-04-02
> >
> > Thank you for your clear responses. All of my questions have been thoroughly addressed.

---

### Decision · Program_Chairs · 2026-04-30

**Decision:**

Accept (spotlight)

**Comment:**

This paper addresses causal discovery from observational count data in the presence of latent confounders, a challenging problem with applications in e commerce, epidemiology, and other domains where branching processes generate discrete data. The authors extend the Poisson Branching Structural Causal Model (PB SCM) to the Latent Confounded PB SCM (LC PB SCM), relaxing the causal sufficiency assumption. The key insight is to map causal mechanisms to the algebraic properties of probability generating functions (PGFs) via a novel “Trie” representation. The paper establishes complete identifiability conditions for local 3 variable substructures (up to monomial equivalence) and translates these theoretical results into a practical algorithm. Experiments on synthetic and real world data demonstrate that the method significantly outperforms existing approaches (PB SCM, FCI, GES, OCD) under latent confounding.

The paper makes a clear theoretical contribution with novel algebraic techniques, and the rebuttal convincingly addressed the concerns about numerical stability, scalability, error propagation, presentation clarity.

Suggested camera ready-reversions include:
* The corrected definitions and typos (as promised).
* The complexity analysis and expanded related work discussion.
* The additional experimental results on larger graphs (12 observed + 8 latent).
* A clear statement of the independent latent assumption as a limitation, with a brief discussion of potential extensions.